# Sequential forward and reverse transport of the Na$^+$ Ca$^{2+}$ exchanger generates Ca$^{2+}$ oscillations within mitochondria

Krishna Samanta[1], Gary R. Mirams[2] & Anant B. Parekh[1]

Mitochondrial Ca$^{2+}$ homoeostasis regulates aerobic metabolism and cell survival. Ca$^{2+}$ flux into mitochondria is mediated by the mitochondrial calcium uniporter (MCU) channel whereas Ca$^{2+}$ export is often through an electrogenic Na$^+$–Ca$^{2+}$ exchanger. Here, we report remarkable functional versatility in mitochondrial Na$^+$–Ca$^{2+}$ exchange under conditions where mitochondria are depolarised. Following physiological stimulation of cell-surface receptors, mitochondrial Na$^+$–Ca$^{2+}$ exchange initially operates in reverse mode, transporting cytosolic Ca$^{2+}$ into the matrix. As matrix Ca$^{2+}$ rises, the exchanger reverts to its forward mode state, extruding Ca$^{2+}$. Transitions between reverse and forward modes generate repetitive oscillations in matrix Ca$^{2+}$. We further show that reverse mode Na$^+$–Ca$^{2+}$ activity is regulated by the mitochondrial fusion protein mitofusin 2. Our results demonstrate that reversible switching between transport modes of an ion exchanger molecule generates functionally relevant oscillations in the levels of the universal Ca$^{2+}$ messenger within an organelle.

---

[1] Department of Physiology, Anatomy and Genetics, Oxford University, Parks Road, Oxford OX1 3PT, UK. [2] Centre for Mathematical Medicine and Biology, School of Mathematical Sciences, Nottingham University, Nottingham NG7 2RD, UK. Correspondence and requests for materials should be addressed to A.B.P. (email: anant.parekh@dpag.ox.ac.uk)

Mitochondrial $Ca^{2+}$ import shapes the pattern of cytosolic $Ca^{2+}$ signals and regulates ATP production and cell survival[1]. The porous outer mitochondrial membrane is freely permeable to $Ca^{2+}$ but the inner membrane is not and therefore requires transporters to shuttle $Ca^{2+}$ between the cytosol and mitochondrial matrix[2]. A major route for mitochondrial $Ca^{2+}$ uptake is through the mitochondrial $Ca^{2+}$ uniporter (MCU), a highly $Ca^{2+}$-selective low conductance ion channel[3,4]. MCU is part of a larger complex involving regulators MICU1 and MICU2, MCUR1 and EMRE[5]. Flux through the MCU complex is determined by the prevailing electrochemical $Ca^{2+}$ gradient[6], with a major factor being the large electrical driving force that arises from the negative potential ($\sim -200$ mV) across the inner mitochondrial membrane.

$Ca^{2+}$ transporters that extrude $Ca^{2+}$ from the matrix have also been characterised at a molecular level and include Letm1 (leucine zipper-EF-hand-containing transmembrane protein 1)[7] and mitochondrial $Na^+-Ca^{2+}$ exchange (NCLX)[8]. Letm1 is a $Ca^{2+}/2H^+$ electroneutral antiporter whereas NCLX is thought to be electrogenic[9], although the precise $Na^+:Ca^{2+}$ stoichiometry is unclear[10]. In one study where the relative contributions of Letm1 and NCLX to mitochondrial $Ca^{2+}$ export was investigated, NCLX was found to play the dominant role[11].

Mitochondria are dynamic organelles, undergoing fusion and fission with the capacity to form reticular networks[12]. The precise architecture of mitochondria is important for cell viability, growth, proliferation and signalling[13]. Mitochondrial fusion is regulated by dynamin-related protein (Drp1) where outer and inner mitochondrial membrane fusion depend on mitofusin 1 and mitofusin 2, and OPA1, respectively[12]. Mitofusin 2 is also found on the endoplasmic/sarcoplasmic reticulum surface and is therefore thought to act as a physical tether bringing mitochondria and endoplasmic/sarcoplasmic reticulum together at specialised regions called mitochondrial associated membranes[14]. Close apposition of the two organelles allows for rapid and effective local $Ca^{2+}$ signalling[15,16]. $Ca^{2+}$ release from the endoplasmic reticulum by the $Ca^{2+}$-releasing second messenger inositol trisphosphate ($InsP_3$) leads to a high local $Ca^{2+}$ signal that can be transported into mitochondria by the MCU. The rise in matrix $Ca^{2+}$ stimulates rate-limiting enzymes in the Krebs cycle[17], resulting in accelerated ATP production. In the heart, for example, mitochondrial fusion dynamics depends on contractile activity[18]. In cardiac myocytes, shuttling of $Ca^{2+}$ released from the sarcoplasmic reticulum by ryanodine receptors into mitochondria drives rapid bioenergetic responses that are important for cardiac function[19].

Stimulation of Gq protein-coupled receptors activate phospholipase C to generate $InsP_3$ (ref. [20]). Low concentrations of agonist, which are thought to mimic physiologically relevant doses, typically evoke oscillations in cytosolic $Ca^{2+}$. The oscillations arise from regenerative $Ca^{2+}$ release from the endoplasmic reticulum by $InsP_3$-gated $Ca^{2+}$ channels followed by store-operated $Ca^{2+}$ entry[20]. Previous work has shown cytosolic $Ca^{2+}$ oscillations following stimulation of native cysteinyl leukotriene type I receptors in mast cells with the natural agonist leukotriene $C_4$ are propagated rapidly and faithfully into mitochondria to generate oscillations in matrix $Ca^{2+}$ (ref. [21]). Knockdown of the MCU or mitochondrial depolarisation, which impairs $Ca^{2+}$ flux through the MCU, suppressed mitochondrial $Ca^{2+}$ uptake[21]. Loss of mitochondrial $Ca^{2+}$ buffering resulted in rundown of cytosolic $Ca^{2+}$ oscillations, which arose through enhanced $Ca^{2+}$-dependent inactivation of $InsP_3$ receptors.

Cytosolic $Ca^{2+}$ oscillations are sustained by $Ca^{2+}$ entry through store-operated $Ca^{2+}$ channels, which refill the endoplasmic reticulum with $Ca^{2+}$ following $InsP_3$-evoked $Ca^{2+}$ release[20]. In mast cells and T lymphocytes, mitochondrial $Ca^{2+}$ uptake sustains $Ca^2$

$^+$ entry by reducing $Ca^{2+}$-dependent slow inactivation of the store-operated $Ca^{2+}$ channels[22,23]. In addition, mitochondria also regulate the redistribution of STIM1, a molecule necessary for the activation of store-operated $Ca^{2+}$ channels, from the endoplasmic reticulum to the plasma membrane[24]. In mast cells, inhibition of store-operated $Ca^{2+}$ influx following mitochondrial depolarisation can be rescued by knockdown of the mitochondrial fusion protein mitofusin 2 (ref. [24]). In this study, we show that mitochondrial $Ca^{2+}$ oscillations induced by leukotriene receptor stimulation that are lost following mitochondrial depolarisation can be rescued when mitofusin 2 levels are reduced. We find that the mitochondrial $Ca^{2+}$ oscillations under these conditions arise from mitochondrial $Na^+-Ca^{2+}$ exchange, operating sequentially in reverse and forward transport modes. Our data show that the same transport molecule can both raise and lower matrix $Ca^{2+}$ in response to receptor stimulation. More generally, our results reveal that an ion transporter can compensate for impaired activity of an ion channel and thereby sustain functionally relevant $Ca^{2+}$ signals within an organelle.

## Results

**Mitochondrial depolarisation impairs $Ca^{2+}$ signals to agonist.** Stimulation of native G-protein-coupled cysteinyl leukotriene type I in RBL-1 mast cells receptors with the agonist leukotriene $C_4$ ($LTC_4$) increases the levels of the second messenger $InsP_3$, which triggers oscillations in cytosolic $Ca^{2+}$ (Fig. 1a)[25]. The oscillations decrease somewhat in number (Fig. 1c) and amplitude (Fig. 1d) over a 600 s period, due to receptor desensitisation[26]. Measurements of matrix $Ca^{2+}$, using ratiometric pericam that is genetically targeted to the mitochondrial matrix, revealed oscillations in matrix $Ca^{2+}$ following $LTC_4$ challenge (Fig. 1e, g, h) that closely mirrored those in the cytosol (Fig. 1a). Mitochondrial depolarisation with the protonophore carbonilcyanide p-triflouromethoxyphenylhydrazone (FCCP) abolished prolonged oscillatory $Ca^{2+}$ signals both in the cytosol (Fig. 1a, c, d) and mitochondrial matrix (Fig. 1e, g, h)[21]. Similar results were obtained after knockdown of the MCU[21].

**Knockdown of Mitofusin 2 rescues matrix $Ca^{2+}$ signals.** siRNA-directed knockdown of mitofusin 2, a dynamin-related GTPase protein involved in mitochondrial fusion[27,28], had no effect on $LTC_4$-evoked oscillations in cytosolic (Fig. 1b–d) or matrix $Ca^{2+}$ (Fig. 1f–h) in cells with an intact mitochondrial membrane potential. However, knockdown of mitofusin 2 fully prevented the loss of cytosolic (Fig. 1b–d) and matrix $Ca^{2+}$ oscillations (Fig. 1f–h) that occurred after mitochondrial depolarisation with FCCP and oligomycin. Western blots confirmed that knockdown of MFN2 led to a significant decrease in protein levels of ~80% (Fig. 1i). Qualitatively similar results were obtained in HEK293 cells expressing the cysteinyl leukotriene type I receptor (Supplementary Figures 1 and 2).

We considered the possibility that the mitochondrial membrane potential repolarised in the presence of FCCP when leukotriene receptors were activated in mitofusin 2-deficient cells. However, this was not the case. Following treatment with FCCP and oligomycin, the potential (measured with TMRE) depolarised and remained low after stimulation with $LTC_4$ for the duration of agonist exposure, both in the presence of mitofusin 2 (control, Fig. 1j) or after knockdown (Fig. 1k).

Mitofusin 2 plays an important role in regulating mitochondrial fusion[29]. However, the rescue of cytosolic $Ca^{2+}$ oscillations that was achieved by knockdown of mitofusin 2 in cells with depolarised mitochondria (Fig. 1b–d) was not mimicked by knockdown of another fusion protein, Optic atrophy 1 (OPA1[13]

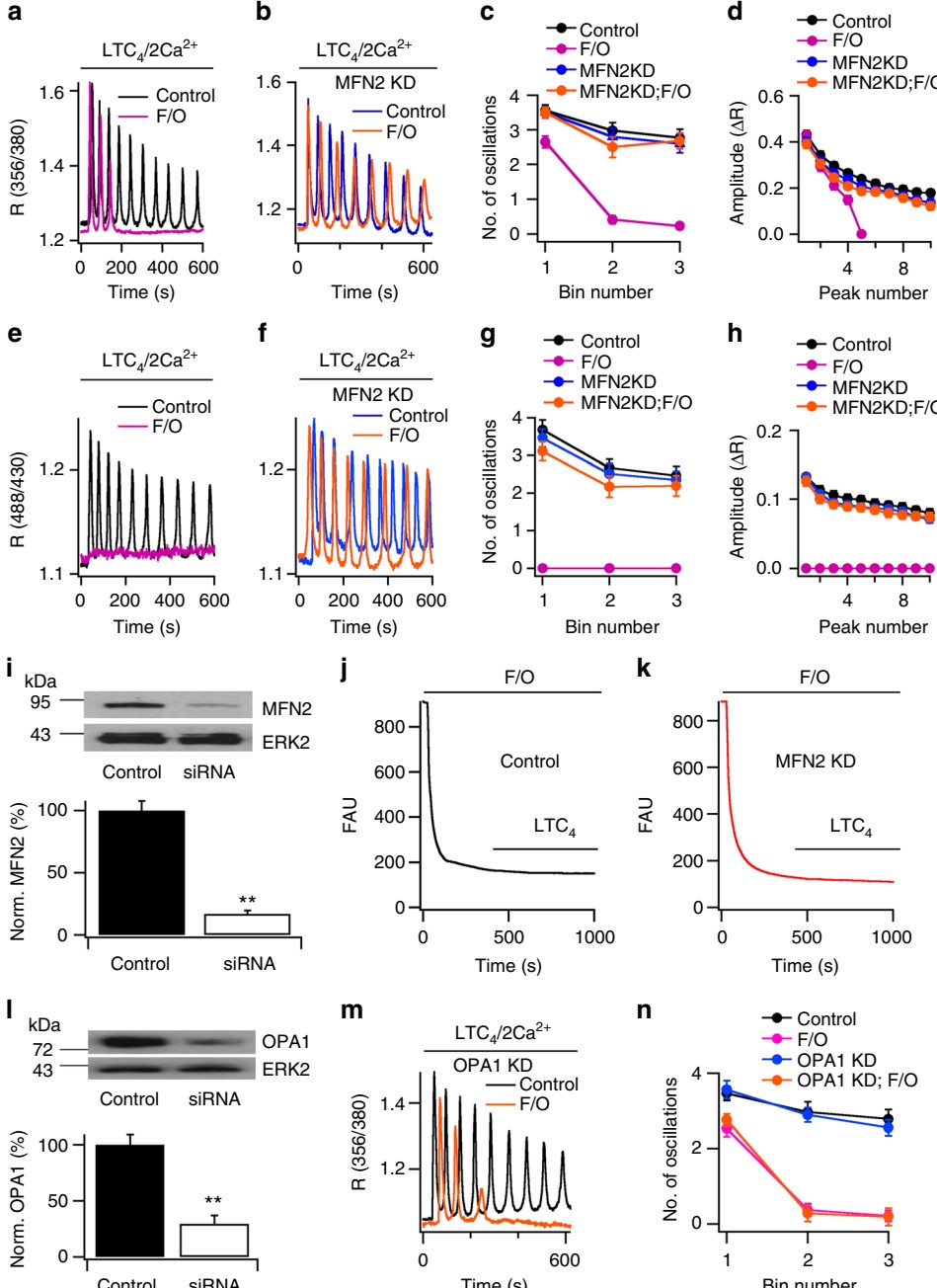

**Fig. 1** Mitofusin 2 knockdown rescues oscillatory $Ca^{2+}$ signals in cells with de-energised mitochondria. **a** Cytosolic $Ca^{2+}$ oscillations (measured with fura 2) to $LTC_4$ (120 nM) are suppressed by FCCP (2 µM) and oligomycin (0.5 µg per ml; depicted as F/O), pre-treated for 5–10 min. **b** Mitofusin 2 knockdown has little effect on cytosolic $Ca^{2+}$ oscillations in a cell with energised mitochondria but sustains the response in a cell pre-treated with FCCP and oligomycin. **c** The number of oscillations in each 200 s bin (measured from the application of $LTC_4$) are compared for the conditions shown. Each point is the mean of 15–20 cells from three independent experiments. Control trace (black) is offset upwards by 0.1, to resolve it from other traces. **d** Oscillation amplitude is plotted against oscillation number. A peak number of 4 denotes the fourth oscillation. Control trace (black) is offset upwards by 0.04. **e** Matrix $Ca^{2+}$ oscillations measured with the mitochondrially targeted pericam are shown for a control cell challenged with $LTC_4$ and for one pre-treated with FCCP and oligomycin. **f** Matrix $Ca^{2+}$ oscillations are compared in the absence and presence of FCCP, following mitofusin 2 knockdown. **g, h** The number of matrix $Ca^{2+}$ oscillations (**g**) and the amplitude of each oscillation (**h**) are compared for the conditions shown. Each point is the mean of >14 cells from three independent experiments. In panel **g**, the control trace (black) is offset upwards by 0.1. **i** Western blot compares mitofusin 2 expression in control cells and after siRNA-targeted knockdown. The histogram summarises aggregate data from two separate samples. **j, k** Mitochondrial membrane potential, measured with TMRE, is shown following exposure first to FCCP and oligomycin and then $LTC_4$ in a control cell (**j**) and in one following mitofusin 2 knockdown (**k**). FAU denotes fluorescence, arbitrary units. **l** Western blot compares OPA1 protein expression in control cells and after siRNA-targeted knockdown. **m** OPA1 knockdown does not rescue cytosolic $Ca^{2+}$ oscillations in FCCP-treated cells. **n** Number of $Ca^{2+}$ oscillations per 200 s bin are compared for the conditions shown. All data are from RBL-1 mast cells. **$p < 0.01$ (unpaired Student's $t$-test). Error bars denote SEM

(Fig. 1l, n). Hence the involvement of mitofusin 2 on matrix $Ca^{2+}$ is distinct from its well-documented role in mitochondrial fusion.

**NCLX rescues matrix $Ca^{2+}$ signals in depolarised mitochondria.** $Ca^{2+}$ flux through MCU is steeply dependent on the voltage gradient across the inner mitochondrial membrane[6] and MCU open probability decreases as the membrane potential depolarises[6]. Depolarisation with FCCP almost fully suppressed mitochondrial $Ca^{2+}$ uptake following receptor stimulation (Fig. 1e), demonstrating little flux through the MCU. The findings that the amplitude and frequency of mitochondrial $Ca^{2+}$ oscillations induced by leukotriene receptor stimulation were similar between control cells, where $Ca^{2+}$ entry into the matrix is provided by the MCU, and in mitofusin 2-deficient cells with depolarised mitochondria, where flux through the MCU had been compromised (Fig. 1g, h), suggest an alterative $Ca^{2+}$ influx pathway operates under conditions of a depolarised mitochondrial membrane potential. A major route for $Ca^{2+}$ efflux from the matrix is through electrogenic $Na^+$–$Ca^{2+}$ exchange in the inner mitochondrial membrane. In energised mitochondria and with resting cytosolic and matrix $Ca^{2+}$ and $Na^+$ concentrations, the exchanger operates in forward mode, exporting $Ca^{2+}$ from the matrix in exchange for cytosolic $Na^+$. However, in depolarised mitochondria, the exchanger switches to reverse mode importing $Ca^{2+}$ into and extruding $Na^+$ from the matrix[9]. We designed experiments to test whether reverse mode $Na^+$–$Ca^{2+}$ exchange provided a route for mitochondrial $Ca^{2+}$ uptake following receptor stimulation in cells with depolarised mitochondria. After knockdown of mitofusin 2, cells were treated acutely with FCCP and oligomycin and then leukotriene receptors were activated while matrix or cytosolic $Ca^{2+}$ was measured. Agonist-evoked cytosolic $Ca^{2+}$ oscillations (Fig. 1b–d) were rapidly propagated into the mitochondrial matrix (Fig. 1f–h). Three independent lines of evidence suggest that mitochondrial $Na^+$–$Ca^{2+}$ exchange is the main route for $Ca^{2+}$ import under these depolarised conditions. First, inhibition of the mitochondrial $Na^+$–$Ca^{2+}$ exchanger with the benzothiazepine CGP-37157 abolished mitochondrial $Ca^{2+}$ uptake following leukotriene receptor activation (Fig. 2a). Second, elevation of cytosolic $Na^+$ concentration should reduce the trans-mitochondrial $Na^+$ gradient and this will decrease reverse mode $Na^+$–$Ca^{2+}$ exchange activity. We raised cytosolic $Na^+$ levels by incubating cells with ouabain, an inhibitor of the plasma membrane $Na^+$–$K^+$ ATPase pump. In the presence of ouabain, mitochondrial $Ca^{2+}$ oscillations were abolished following stimulation with LTC$_4$ in mitofusin 2-deficient cells with depolarised mitochondria (Fig. 2b). Three patterns of response were observed after ouabain treatment (Fig. 2b, c): some cells (45%) failed to respond at all to LTC$_4$, others did so by giving a modest rise in matrix $Ca^{2+}$ (30%) and some responded by generating 1 or more small matrix $Ca^{2+}$ spikes (25%). Ouabain had no inhibitory effect when cells were stimulated with agonist in the absence of FCCP and oligomycin (Supplementary Figure 3). Thirdly, we used an siRNA approach to knock down the recently discovered mitochondrial $Na^+$–$Ca^{2+}$ exchanger[8]. Significant knockdown of the exchanger was seen in western blots (Fig. 2d; knockdown was $68.3 \pm 7.2\%$ in two independent experiments). In all siRNA-treated cells, mitochondrial $Ca^{2+}$ uptake was reduced but to differing extents (Fig. 2e). In some cells (labelled blank in Fig. 2f), $Ca^{2+}$ import was almost completely suppressed. In others (labelled small response in Fig. 2f), a single matrix $Ca^{2+}$ spike occurred whereas in a third group (labelled >1 spike), a couple of small $Ca^{2+}$ oscillations developed followed by a quiescent period and then one or two oscillations reappeared several tens of seconds later (Fig. 2e, f). Despite the variability, each group was very different from the corresponding control, which showed

repetitive $Ca^{2+}$ oscillations for several tens of seconds. Collectively, these experiments reveal a major role for mitochondrial $Na^+$–$Ca^{2+}$ exchange in driving oscillations in matrix $Ca^{2+}$ following stimulation of cell-surface receptors.

To see whether the oscillations in matrix $Ca^{2+}$ indeed reflected transport through $Na^+$–$Ca^{2+}$ exchange, we expressed a catalytically inactive mutant in which threonine 468 was replaced by serine[8]. In these experiments, endogenous exchanger protein was first knocked down using siRNA and then either wild type or mutant exchanger plasmid transfected 24 h later. Matrix $Ca^{2+}$ oscillations to LTC$_4$ stimulation were seen in ~50% (11/20) of FCCP-treated cells overexpressing $Na^+$–$Ca^{2+}$ exchange but no responses (21/21 cells) were observed when the catalytic mutant was expressed instead (Fig. 2g).

We considered other possibilities that could account for the increased mitochondrial $Ca^{2+}$ uptake in FCCP-treated cells lacking mitofusin 2. In mitofusin 2-knock out mouse embryonic fibroblasts, MCU levels decrease by ~50%[30]. In HEK293 cells, transient knockdown of mitofusin 2 had no significant effect on MCU expression (Fig. 2h; measured using quantitative PCR 24 h after mitousin 2 knockdown), suggesting alterations in MCU expression are unlikely to explain the rescue of matrix $Ca^{2+}$ signals in depolarised mitochondria. Dissipation of the mitochondrial potential can open the large conductance permeability transition pore, providing a route for $Ca^{2+}$ transport across the inner mitochondrial membrane. However, the permeability transition pore inhibitor cyclosporine A failed to affect agonist-evoked oscillations in matrix $Ca^{2+}$ in mitofusin 2-deficient cells in the presence of FCCP (Fig. 2i; $10.3 \pm 0.2$ oscillations were generated over 600 s in control cells and the corresponding value in cyclosporine A-treated cells was $10.1 \pm 0.2$). FCCP and oligomycin treatment could lower cytosolic pH and this might explain why agonist-evoked $Ca^{2+}$ signals are impaired. However, FCCP and oligomycin had little effect on cytosolic pH (Fig. 2j).

**Functional coupling between MCU and NCLX.** We asked whether LTC$_4$-induced matrix $Ca^{2+}$ oscillations in FCCP-treated cells lacking mitofusin 2 were totally independent of the MCU. Following knock down of both the MCU and mitofusin 2, stimulation with agonist in cells with a depolarised mitochondrial membrane potential now consistently failed to generate oscillations in matrix $Ca^{2+}$ (Supplementary Figure 4). We hypothesised that, despite providing an exceedingly small $Ca^{2+}$ flux into the matrix under the depolarised conditions that occur in the presence of FCCP[6], the MCU nevertheless provided either trigger or facilitory matrix $Ca^{2+}$ for driving $Na^+$–$Ca^{2+}$ exchange activity. To test this, we raised matrix $Ca^{2+}$ very slightly by applying a low dose of the $Ca^{2+}$ ionophore ionomycin (2 nM) to intact cells. In cells in which we reduced both MCU and mitofusin 2 expression and then depolarised mitochondria, oscillations in matrix $Ca^{2+}$ to leukotriene receptor stimulation were rescued only if matrix $Ca^{2+}$ had been increased slightly by ionomycin prior to agonist exposure (Fig. 3a; aggregate data are summarised in Fig. 3c). The small increase in matrix $Ca^{2+}$ induced by ionomycin per se did not trigger $Ca^{2+}$ oscillations (Fig. 3a). The oscillations in matrix $Ca^{2+}$ were suppressed by CGP-37157, confirming they were mediated through $Na^+$–$Ca^{2+}$ exchange (Fig. 3a). Although LTC$_4$ failed to elicit matrix $Ca^{2+}$ oscillations in MCU-deficient cells, subsequent application of ionomycin rescued the oscillatory response and this was blocked by CGP-37157 (Fig. 3b; aggregate data are summarised in Fig. 3c).

**NCLX activity in permeabilised cells.** To trap mitochondrial $Na^+$–$Ca^{2+}$ exchange in forward or reverse modes, we clamped cytosolic $Na^+$ and $Ca^{2+}$ at fixed concentrations using digitonin-

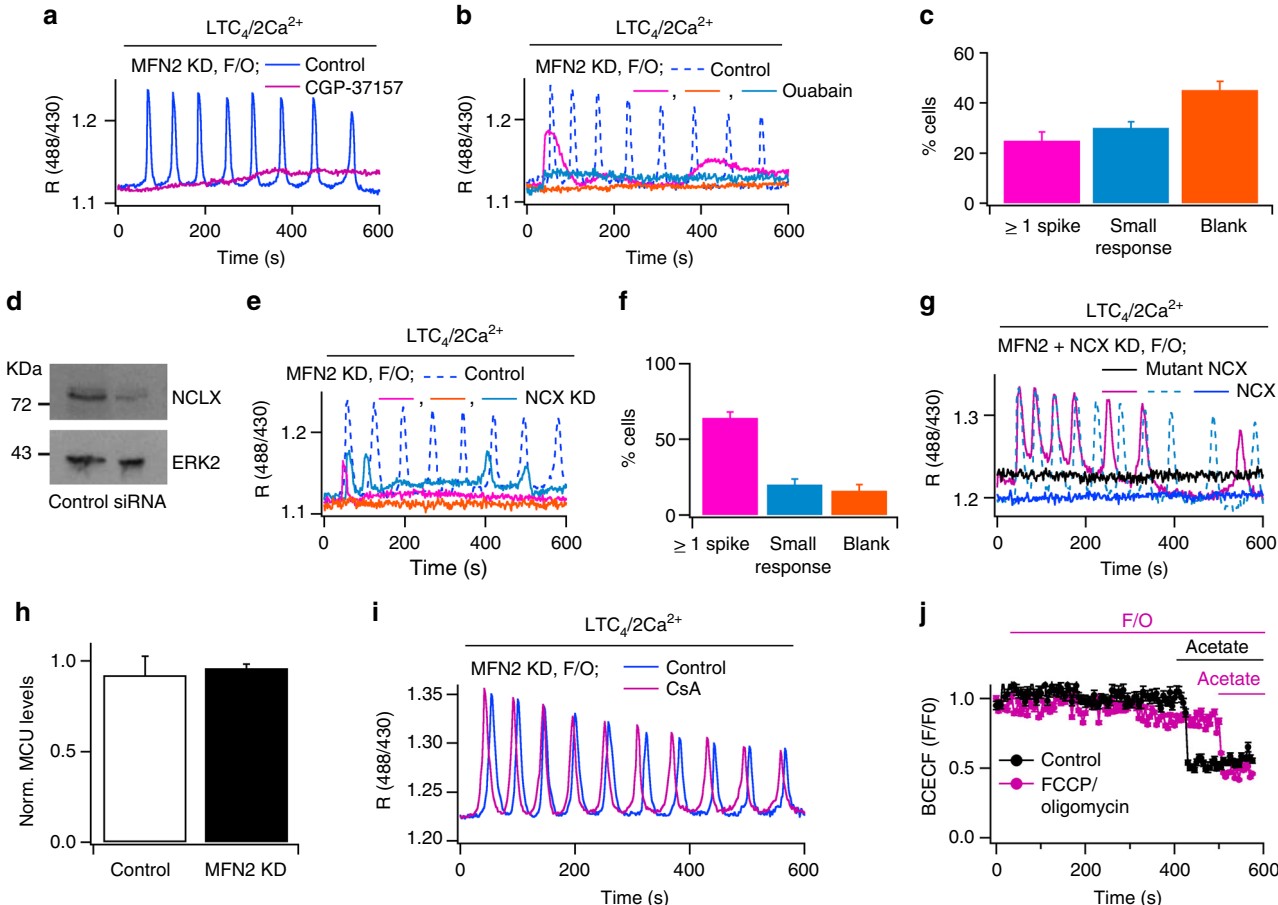

**Fig. 2** Oscillations in matrix $Ca^{2+}$ by mitochondrial $Na^+$–$Ca^{2+}$ exchange. In panels **a**–**f**, mitofusin 2 was knocked down and cells were pre-treated acutely with FCCP and oligomycin prior to $LTC_4$ challenge. Matrix $Ca^{2+}$ was measured with ratiometric pericam. **a** Pre-exposure (10 min) to CGP-37157 (10 μM) abolished matrix $Ca^{2+}$ oscillations to $LTC_4$ (control: blue trace, typical of 21/29 cells; CGP-treated: red trace, typical of 12/12 cells). **b** As in panel **a**, but ouabain (25 μM; 15 min pre-treatment) was used to raise cytosolic $Na^+$. Three types of matrix $Ca^{2+}$ response to $LTC_4$ were now observed; blank/no response (orange), a small slow rise (blue) or at least one spike (magenta). **c** Bar plot shows the fractional responses, as in panel **b**. Total number of cells analysed was 24. **d** Western blot compares $Na^+$–$Ca^{2+}$ exchanger expression in control cells and in cells after siRNA-directed knockdown. **e** $Ca^{2+}$ oscillations to $LTC_4$ are suppressed following $Na^+$–$Ca^{2+}$ exchanger knockdown. Three types of response were now observed (see panel **f**). **f** Bar plot shows the types of responses of matrix $Ca^{2+}$ to $LTC_4$ following exchanger knockdown (as described in panel **c**). Total number of cells analysed was 20. **g** Matrix $Ca^{2+}$ oscillations to $LTC_4$ were induced in control cells but not in those expressing a catalytic mutant of the exchanger. Two types of response were seen when the wild type exchanger was overexpressed: no response (9/20 cells) and oscillatory responses (11/20 cells). The two oscillatory responses (dashed blue and magenta traces) depict the two patterns of oscillatory response. **h** Histogram compares MCU expression before and after mitofusin 2 had been knocked down. **i** Matrix $Ca^{2+}$ oscillations to $LTC_4$ are compared between a control cell and one pre-exposed (15 min) to cyclosporine A. In this experiment, mitofusin 2 was knocked down and FCCP/oligomycin applied 8 min before stimulation. **j** Cytosolic pH, measured with BCECF, is compared for control cells (black trace) and cells acutely exposed to FCCP/oligomycin (as indicated; red trace). Acetate was applied at the end of the experiment to induce cytosolic acidification. Each trace is the mean of between 16 and 21 cells. Error bars denote SEM

permeabilised cells[9]. $Ca^{2+}$ released from the endoplasmic reticulum by application of exogenous $InsP_3$ in permeabilised RBL cells is rapidly taken up into mitochondria by the MCU[31,32]. In control cells with 10 mM cytosolic $Na^+$, mitochondria sequestrated $Ca^{2+}$ that had been released by $InsP_3$ to produce a sustained elevation in matrix $Ca^{2+}$ and this was prevented by pre-treatment with FCCP and oligomycin (Supplementary Figure 5). The matrix $Ca^{2+}$ rise in response to $InsP_3$ challenge was also prevented by pre-incubation with the $InsP_3$ receptor antagonist heparin or if stores had been depleted of $Ca^{2+}$ by prior exposure to thapsigargin (Supplementary Figure 5), demonstrating the matrix $Ca^{2+}$ rise is due to $InsP_3$-dependent $Ca^{2+}$ release from the endoplasmic reticulum. After knockdown of MCU and MFN2, we raised matrix $Ca^{2+}$ slightly by exposure to 2 nM ionomycin and then applied $InsP_3$. $InsP_3$ now failed to raise matrix $Ca^{2+}$ (red trace in Fig. 3d; aggregate data in Fig. 3f). However, if mitochondria were

depolarised by exposure to FCCP in MCU- and mitofusin 2-deficient cells, matrix $Ca^{2+}$ increased following $InsP_3$ stimulation (Fig. 3d, f), indicating reverse mode of the exchanger. This increase in matrix $Ca^{2+}$ was sustained for several seconds in the continued presence of $InsP_3$ but fell rapidly when cytosolic $Ca^{2+}$ was lowered by perfusion with the $Ca^{2+}$ chelator BAPTA (Fig. 3d; decay half-time of $18.6 \pm 2.0\,s$), indicating the $Na^+$–$Ca^{2+}$ exchanger was now in forward mode, transporting $Na^+$ into the matrix in exchange for matrix $Ca^{2+}$. CGP-37157 prevented the $InsP_3$-induced rise in matrix $Ca^{2+}$ in depolarised mitochondria (Fig. 3d).

In MFN2- and MCU-deficient permeabilised cells bathed in low (1 mM) $Na^+$-solution and FCCP and oligomycin to trap the exchanger in reverse mode, matrix $Ca^{2+}$ rose upon exposure to $InsP_3$ (Fig. 3e; aggregate data are shown in Fig. 3f) and remained elevated, even after addition of BAPTA to the cytosol (Fig. 3e;

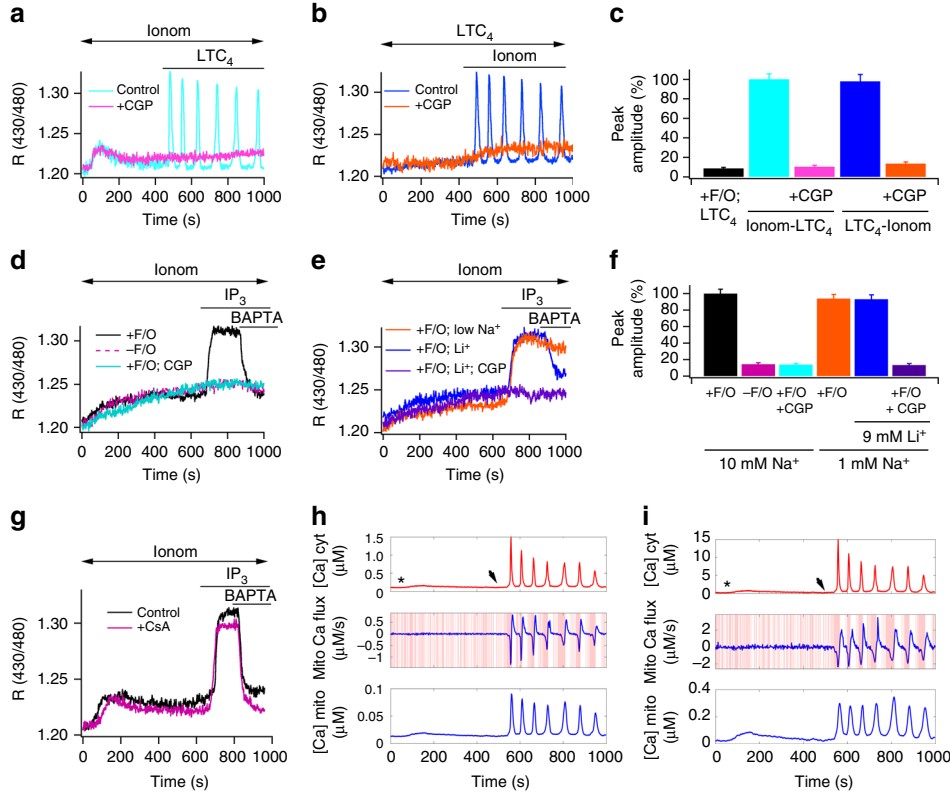

**Fig. 3** Forward and reverse mode Na$^+$–Ca$^{2+}$ exchanger. **a** After knockdown of MCU and MFN2, matrix Ca$^{2+}$ oscillations are induced by LTC$_4$ in FCCP-treated cells following a small rise in matrix Ca$^{2+}$ by 2 nM ionomycin. Oscillations were seen in 15/24 cells. CGP-37157 suppressed this response (15/15 cells). **b** As in panel **a** but LTC$_4$ is now applied first. Oscillations were seen in 10/15 cells. CGP-37157 blocked the response in 18/18 cells. **c** Aggregate data are compared. All groups were pre-treated with FCCP and oligomycin and both MCU and MFN2 were knocked down. Ionom-LTC$_4$ denotes ionomycin followed by LTC$_4$ (as in panel **a**). LTC$_4$-Ionom denotes LTC$_4$ then ionomycin, as in panel **b**. **d** Matrix Ca$^{2+}$ is measured in permeabilised cells following stimulation with InsP$_3$. Cytosolic Na$^+$ was 10 mM and cytosolic Ca$^{2+}$ was weakly buffered at 200 nM. Ionomycin was applied to raise matrix Ca$^{2+}$, then InsP$_3$ was added. F/O denotes FCCP/oligomycin. BAPTA was perfused to rapidly reduce cytosolic Ca$^{2+}$. **e** As in panel **d**, but cytosolic Na$^+$ was 1 mM. In the experiments with Li$^+$, 9 mM Li$^+$ was added to the 1 mM Na$^+$ solution. **f** Aggregate data from experiments as in panels **d** and **e** are compared. Each bar denotes between 15 and 23 cells. **g** Ca$^{2+}$ signals in permeabilised cells are compared in the absence and presence of cyclosporine A (1 μM). Here, mitofusin 2 had been knocked down and FCCP/oligomycin added 5–10 min before simulation with InsP$_3$. Control trace is in the absence of cyclosporine A. For both traces, 1 μM ruthenium red was present throughout. **h** Simulations of Na$^+$–Ca$^{2+}$ exchanger flux (middle panel, with pink highlighted background showing where NCX is operating in reverse mode) and matrix free Ca$^{2+}$ (bottom panel) following a cytosolic Ca$^{2+}$ rise (upper panel) obtained from fura 2-loaded intact cells. Cells were exposed to 2 nM ionomycin (asterisk) before LTC$_4$ challenge (marked by arrow; same protocol as in panel **a**). The cytosolic Ca$^{2+}$ concentration indicated is the measured bulk concentration. **i** As in panel **h** but cytosolic Ca$^{2+}$ has been estimated to reach 15 μM (see text). Error bars denote SEM

decay half-time of 108.2 ± 4.8 s). Because Li$^+$ can partially replace Na$^+$ in the transport cycles of the exchanger[8], we added 9 mM Li$^+$ to the low Na$^+$ cytosolic solution. Stimulation with InsP$_3$ led to a rise in matrix Ca$^{2+}$ but perfusion with BAPTA now reduced matrix Ca$^{2+}$ (Fig. 3e; decay half-time of 30.2 ± 2.7 s). The matrix Ca$^{2+}$ rise induced by InsP$_3$ in the presence of Li$^+$ was prevented by CGP-37157 (Fig. 3e; aggregate data are shown in Fig. 3f). Collectively, these data are consistent with a major role for the exchanger operating in reverse mode to raise matrix Ca$^{2+}$ and then in forward mode to lower it.

To inhibit MCU fully, we used the permeabilised cell system to apply the membrane-impermeable MCU inhibitor ruthenium red. In control experiments, raising cytosolic Ca$^{2+}$ to 10 μM led to a large rise in matrix Ca$^{2+}$ and this was suppressed by ruthenium red (Supplementary Figure 6). Addition of 2 nM ionomycin in 200 nM Ca$^{2+}$ to ruthenium red- and FCCP-treated cells deficient in mitofusin 2 led to the typical small rise in matrix in Ca$^{2+}$ and InsP$_3$ evoked a further increase in matrix Ca$^{2+}$ (Fig. 3g). This result confirms the existence of a ruthenium red-insensitive Ca$^{2+}$

uptake activated under depolarised conditions. The Ca$^{2+}$ signal decayed to basal levels when BAPTA was subsequently added to the cytosol (Fig. 3g). As was the case in intact cells, cyclosporine A had no effect on the matrix rise induced by InsP$_3$ (Fig. 3g; control pericam ratio increase was 0.076 ± 0.003 and in cyclosporine A it was 0.075 ± 0.002).

**Modelling NCLX activity predicts matrix Ca$^{2+}$ oscillations.** Our findings could be replicated by a mathematical model in which the Na$^+$–Ca$^{2+}$ exchanger was the only functional Ca$^{2+}$ transporter in the inner membrane of depolarised mitochondria (Fig. 3h, i). We fed into the model experimental data of oscillations in cytosolic Ca$^{2+}$ induced by LTC$_4$ following application of 2 nM ionomycin following the protocol used in Fig. 3a, but now obtained from intact cells (Fig. 3h, upper panel). The simulations revealed repetitive fluctuations in exchanger forward and reverse modes (Fig. 3h, middle panel) that led to oscillations in matrix Ca$^{2+}$ (Fig. 3h, lower panel). The oscillatory rise in matrix free

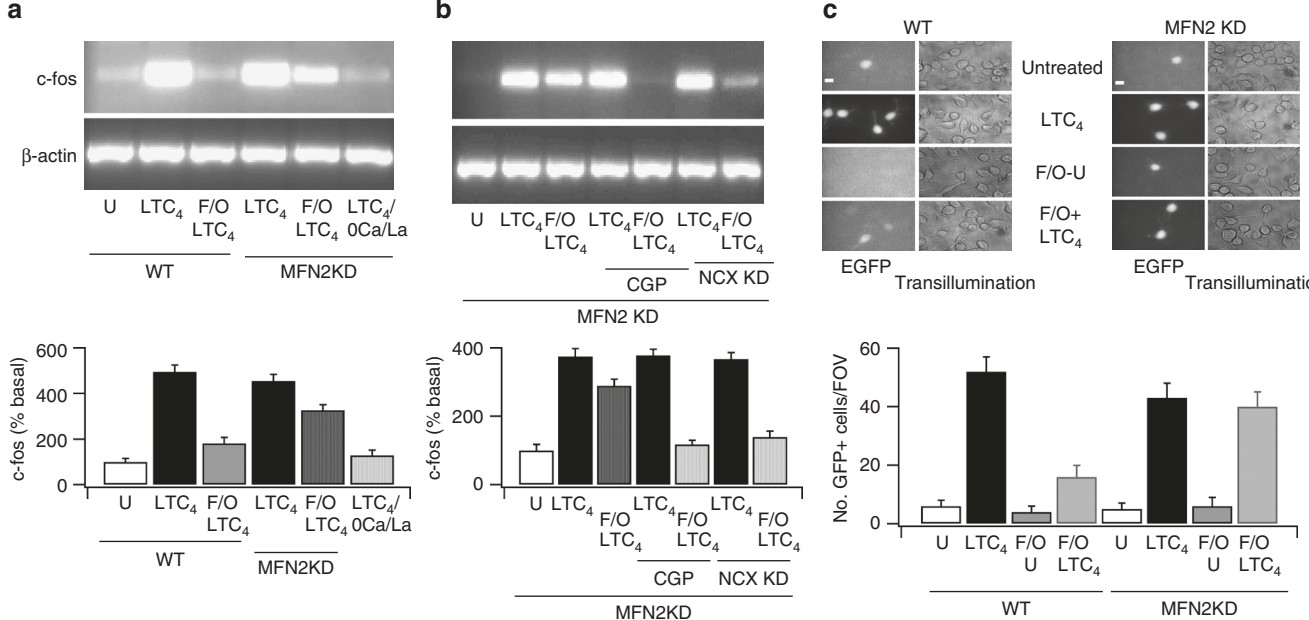

**Fig. 4** Functional Na$^+$–Ca$^{2+}$ exchange supports gene expression in cells with de-energised mitochondria. **a** LTC$_4$-induced c-fos gene transcription is reduced by FCCP but rescued following knockdown of mitofusin 2. Lower panel shows aggregate data from three independent experiments. U denotes untreated, F/O FCCP and oligomycin. F/O was applied 5 min prior to stimulation. LTC4/0Ca/La denotes the lack of c-fos response in the absence of Ca$^{2+}$ influx, despite sustained cytosolic Ca$^{2+}$ oscillations[25]. **b** In mitofusin 2-deficient cells, rescue of LTC$_4$-induced c-fos transcription in the presence of mitochondrial depolarisation is prevented by either CGP-37157 (labelled CGP) or knockdown of the Na$^+$–Ca$^{2+}$ exchanger (NCX KD). **c** NFAT-dependent reporter gene expression is compared for the different conditions shown. The left-hand panel denotes wild-type cells and the right-hand panel responses after knockdown of mitofusin 2. Bar plot in the lower panel denotes aggregate data for the various conditions shown. FOV on y-axis is Field Of View. Here, FCCP and oligomycin were applied 5–10 min before stimulation with LTC$_4$. Agonist and FCCP/oligomycin were removed after a further 8 min. Scale bar denotes 10 μm. Error bars denote SEM

Ca$^{2+}$ concentration depended on the amplitude of the cytosolic Ca$^{2+}$ rise. If we fed in the amplitude of measured bulk cytosolic Ca$^{2+}$ rise then the matrix oscillations were small in size (Fig. 3h). Mitochondria sense high local Ca$^{2+}$ from open InsP$_3$ receptors at specialised regions where the ER is tethered to mitochondria[15,16]. At these sites, mitochondria are exposed to local cytosolic Ca$^{2+}$ concentrations of tens of μM[1,32]. Cytosolic Ca$^{2+}$ signals of this size led to larger amplitude matrix Ca$^{2+}$ fluctuations through forward and reverse mode Na$^+$–Ca$^{2+}$ exchange activity (Fig. 3i). Patch clamp recordings on mitoplasts have reported ~1.5 pA unitary current through the MCU (P$_{open}$ of 0.99) and a whole mitoplast current of ~800 pA, suggesting a single mitoplast expresses ~500 channels[6]. Therefore we simulated matrix Ca$^{2+}$ oscillations for between 1 and 10,000 exchangers. The number of exchangers in Fig. 3h, i were set at 100, with a mitochondrial Ca$^{2+}$ buffering capacity of 100 (Fig. 3h) or 1000 (Fig. 3i). The simulations show qualitatively that forward and reverse mode Na$^+$–Ca$^{2+}$ exchange is sufficient to generate oscillations in matrix Ca$^{2+}$ that replicate those seen experimentally.

**NCLX activity supports Ca$^{2+}$dependent gene expression.** We asked whether mitochondrial Ca$^{2+}$ import via reverse mode Na$^+$–Ca$^{2+}$ exchange was of functional significance. Local Ca$^{2+}$ influx through CRAC channels following leukotriene receptor activation induces Ca$^{2+}$-dependent expression of the immediate early gene c-fos in RBL-1 mast cells[25,33]. Stimulation with LTC$_4$ increased c-fos transcription and this was suppressed by mitochondrial depolarisation following pre-exposure to FCCP and oligomycin for 5 min (Fig. 4a; aggregate data in lower panel), an effect that arises from loss of mitochondrial Ca$^{2+}$ buffering of Ca$^{2+}$ entry through CRAC channels and which then leads to enhanced Ca$^{2+}$-dependent slow inactivation of the channels[22,34]. Knockdown

of mitofusin 2 had no effect on c-fos expression induced by LTC$_4$ but reversed the inhibitory effect of mitochondrial depolarisation (Fig. 4a; aggregate data shown in lower panel). The recovery of c-fos expression in mitofusin 2-deficient cells with depolarised mitochondria was prevented either by exposing cells to CGP-37157 prior to stimulation with LTC$_4$ or knocking down the mitochondrial Na$^+$–Ca$^{2+}$ exchanger (Fig. 4b).

Ca$^{2+}$ microdomains near CRAC channels activated by LTC$_4$ also stimulate the transcription factor NFAT, resulting in expression of a GFP reporter gene driven by an NFAT promoter[35,36]. Stimulation with LTC$_4$ induced a substantial increase in the number of GFP-positive cells (Fig. 4c; aggregate data in lower panel) and this was suppressed by brief pre-treatment with FCCP and oligomycin prior to agonist challenge. Knockdown of mitofusin 2 rescued leukotriene receptor-evoked NFAT-driven gene expression in cells with depolarised mitochondria (Fig. 4c).

## Discussion

Our results show sequential cycling between reverse and forward transport modes of the mitochondrial Na$^+$–Ca$^{2+}$ exchanger results in bidirectional movement of Ca$^{2+}$ across the inner mitochondrial membrane. Bidirectional transport by mitochondrial Na$^+$–Ca$^{2+}$ exchange develops in response to physiological levels of receptor stimulation and provides a mechanism whereby cytosolic Ca$^{2+}$ oscillations can faithfully reconvene within an organelle. Reverse mode transport is regulated by mitofusin 2 and requires depolarisation of the mitochondrial membrane potential. A cartoon depicting this is shown in Fig. 5.

The reversal potential of Na$^+$–Ca$^{2+}$ exchange is determined by the stoichiometry of ion transport and the Nernst potentials for Na$^+$ and Ca$^{2+}$. With an inner mitochondrial membrane potential

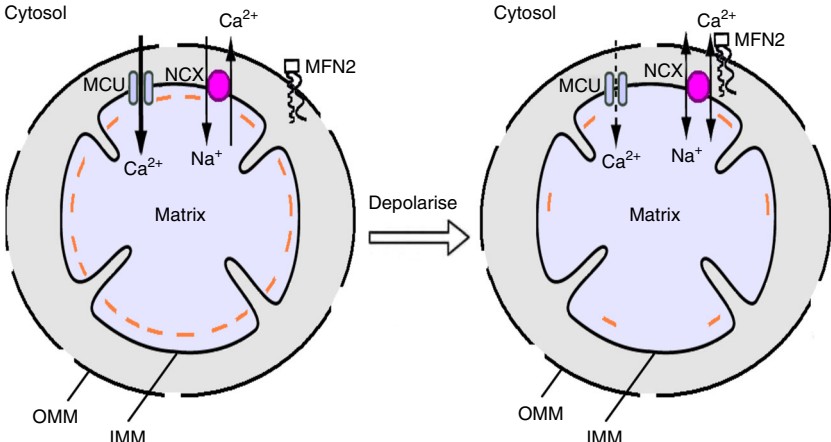

**Fig. 5** Cartoon summarises Na$^+$–Ca$^{2+}$ exchanger transport in mitochondria under the conditions shown The left-hand panel depicts the situation in energised mitochondria. The exchanger operates in forward mode, transporting cytosolic Na$^+$ into the matrix in exchange for matrix Ca$^{2+}$, which has entered through the MCU. Mitofusin 2 is located in the OMM and functionally detached from the exchanger. After mitochondrial depolarisation (right-hand panel), the exchanger is now functionally coupled to mitofusin 2 and can operate in both forward and reverse modes. OMM denotes outer mitochondrial membrane, IMM inner mitochondrial membrane

of ~−180 mV and with typical resting cytosolic and matrix Ca$^{2+}$ and Na$^+$ concentrations, the exchanger will operate in forward mode, transporting Na$^+$ ions from the cytosol into the matrix in exchange for a Ca$^{2+}$ ion. Depolarisation with FCCP enables the exchanger to operate in both forward and reverse modes, depending on the ambient cytosolic Ca$^{2+}$. FCCP induces a depolarisation of ~100 mV[9] and so quite marked depolarisation is needed for the exchanger to operate in reverse mode. Depolarisation on this scale is not observed routinely in living cells and reverse Na$^+$–Ca$^{2+}$ exchange is unlikely to be of physiological significance for the entire mitochondrial pool. However, in pathophysiological states such as oxidative stress induced by ischaemia or strong local reactive oxygen species production, mitochondrial membrane potential in cardiomyocytes can oscillate or even collapse[37,38]. It is important to note that we found reverse mode Na$^+$–Ca$^{2+}$ exchange to be effective only after the combination of mitochondrial depolarisation and knockdown of mitofusin 2. Although mitofusin 2 is widely expressed, it is not expressed uniformly between cells. Muscle has very high levels compared with other cell types. Mitofusin 2 expression is also under control of circulating factors. For example, tumour necrosis factor-α and interleukin-6 both significantly reduce mitofusin 2 expression[39], suggesting the levels expressed are dynamic. Interestingly, expression of mitofusin 2 also alters in various states[39]. Expression falls significantly in skeletal muscle from type 2 diabetes sufferers as well as in obese patients[39]. Hence it is possible that mitofusin 2 levels might fluctuate sufficiently under certain conditions for the protein to regulate Na$^+$–Ca$^{2+}$ exchange activity.

The mechanism whereby mitofusin 2 regulates reverse mode but not forward mode is currently unclear but does not require mitochondrial fusion because knockdown of OPA1 did not replicate the effects of reduction in mitofusin 2. Mitofusin 2 is in the outer mitochondrial membrane and Na$^+$–Ca$^{2+}$ exchange is in the inner membrane. It is possible that the two proteins physically interact after mitochondrial depolarisation and this stabilises or enables the reverse mode to operate. Co-immunoprecipitation studies show that myc-tagged Na$^+$–Ca$^{2+}$ exchanger is present following pulldown of mitofusin 2 and the amount of exchanger increases in the presence of FCCP (Samanta and Parekh, data presented to reviewers). Reverse mode Na$^+$–Ca$^{2+}$ exchanger activity required a small increase in matrix Ca$^{2+}$. This was accomplished by the very low flux through the MCU in depolarised mitochondria or after artificial elevation of matrix Ca$^{2+}$ by a low dose of ionomycin. Whether matrix Ca$^{2+}$ directly binds to the exchanger or its effects are mediated by an intermediary mechanism is currently under investigation.

The mitochondrial Na$^+$–Ca$^{2+}$ exchanger, alternating between transport modes generates oscillations in Ca$^{2+}$ within the mitochondrial matrix and these faithfully reflect the cytosolic Ca$^{2+}$ oscillations that trigger exchanger activity. Although the MCU is the main route for mitochondrial Ca$^{2+}$ uptake, our data reveal that the exchanger can, under certain conditions, provide an alternative route for Ca$^{2+}$ entry. More generally, our data show that forward and reverse mode activities of an ion transporter can substitute for a Ca$^{2+}$ channel in generating Ca$^{2+}$ signals evoked by a physiologically relevant agonist.

## Methods

**Cell culture and transfection**. Rat basophilic leukaemia (RBL-1) and HEK293 cells were purchased from ATCC and were cultured (37 °C, 5% CO$_2$) in Dulbecco's modified Eagle's medium supplemented with 10% fetal bovine serum, 2 mM L-glutamine and penicillin–streptomycin. RBL cells were transfected with the AMAXA system, using nucleofector cell line kit V solution (from Lonza, Cat. No. VCA-1003) and program T-30 were used. Transfection of HEK293 cells was achieved using the lipofectamine method[35]. For transfection of one dish (10 cm), 50 μl of Opti-MEM was mixed with 4 μl of Lipofectamine 2000 (from Invitrogen, Cat. No. 11668-019) in one eppendorf tube and 50 μl of Opti-MEM was mixed with the required amount of DNA or siRNA in another eppendorf tube. The components were gently mixed by pipetting and after 5 min all the components were mixed together. This transfection mixture was then incubated at room temperature for 20 min to generate lipoplexes for transfection. Thereafter, 1 ml of cell culture medium was added to the transfection mixture, which was then pipetted into the dish. After 1 h incubation at 37 °C and 5% CO$_2$, the transfection mixture containing medium was replaced by 2 ml fresh medium. Cells were cultured for 24–48 h in an incubator and then used for experiments.

**Cytosolic Ca$^{2+}$ measurements**. Cytosolic Ca$^{2+}$ was measured using the Ca$^{2+}$-sensitive fluorescent dye fura 2. All experiments were conducted at room temperature, using the IMAGO CCD camera-based system from TILL Photonics[40,41]. Cells were incubated with Fura 2-AM (1 μM) for 40 min at room temperature in the dark and then washed several times in standard external solution of composition (in mM): NaCl 145, KCl 2.8, CaCl$_2$ 2, MgCl$_2$ 2, D-glucose 10, HEPES 10, pH 7.4 with NaOH. After a 15 min de-esterification period, cells were alternately excited at 356 and 380 nm (20-ms exposures) and emission collected >505 nm. Images were acquired every 2 s. Ca$^{2+}$ signals are plotted as R, which denotes the 356/380 nm ratio.

**Cytosolic pH measurements**. Cells were incubated with BCECF-AM (5 μg per ml) for 30 min at room temperature in the dark and then washed three times in standard external solution of composition (in mM): NaCl 145, KCl 2.8, CaCl$_2$ 2,

MgCl$_2$ 2, D-glucose 10, HEPES 10, pH 7.4 with NaOH. Cells were alternately excited at 490 and 440 nm (20-ms exposures) and images were acquired every 5 s. The 490/440 nm ratio was calculated and pH signals are plotted as R/R0, where R0 denotes the resting ratio.

**Mitochondrial Ca$^{2+}$ measurements.** Matrix Ca$^{2+}$ was measured using the genetically encoded ratiometric pericam protein[21]. Following transfection, with pericam, recordings commenced 24 h later using the TILL Photonics system. Cells were illuminated alternatively at 430 and 488 nm (20 ms exposures) at 0.5 Hz and the emitted light was filtered at >510 nm.

**Mitochondrial membrane potential.** Cells were loaded with TMRE (50 nM) in standard external solution for 30 min in the dark, followed by several washes in external solution. Cells were excited at 545 nm and emitted light was collected at >560 nm.

**Cell permeabilisation.** Cells expressing pericam were permeabilised by exposure to 5 μM digitonin for 5 min in an intracellular medium containing: 120 mM KCl, 9 mM NaCl, 2 mM KH$_2$PO$_4$, 10 mM HEPES, 1 mM MgCl$_2$, 1 mM Na-pyruvate, 2 mM Mg-ATP, 50 μM EGTA (pH 7.2). Low Na$^+$-based intracellular medium contained 129 mM KCl, 2 mM KH$_2$PO$_4$, 10 mM HEPES, 1 mM MgCl$_2$, 1 mM Na-pyruate, 2 mM Mg-ATP, 50 μM EGTA (pH 7.2). Li$^+$-based intracellular medium contained: 120 mM KCl, 9 mM LiCl 2 mM KH$_2$PO$_4$, 10 mM HEPES, 1 mM MgCl$_2$, 1 mM Na$^+$-pyruvate, 2 mM Mg-ATP, 50 μM EGTA (pH 7.2). In a separate series of experiments, the exposure time for digitonin was determined experimentally by tracking the loss of the fura 2 fluorescence signal at 356 nm from the cytosol after loading with fura 2-AM, as described for cytosolic Ca$^{2+}$ measurements. Intracellular solutions with different Ca$^{2+}$ concentrations (200 nM; 10 μM) were perfused onto the cells as indicated in the text.

**Western blot.** Total cell lysates were separated on 10% denaturing sodium dodecyl sulfate polyacrylamide gel electrophoresis. Membranes were blocked with 5% nonfat dry milk in phosphate-buffered saline (PBS) plus 0.1% Tween 20 (PBST) buffer for 1 h at room temperature, were washed with PBST three times and then incubated with primary antibody overnight at 4 °C. ERK2 antibody was from Santa Cruz Biotechnology (Cat. No. sc-154) and was used at a dilution of 1:5000. The antibodies against mitofusin 2 and OPA1 were from Santa Cruz Biotechnology (Cat. No. sc-100560) and BD Transduction Laboratories (Cat. No. 612606) respectively, and used at dilutions of 1:250 and 1:1000. NCLX antibody was kindly provided by Prof. Israel Sekler and was used at 1:1000–1:2000 dilution. The membranes were then washed with PBST again and incubated with 1:2500 dilutions of peroxidase-linked anti-rabbit IgG from Santa Cruz Biotechnology (Cat. No. sc-2004) or anti mouse IgG from BD Bioscience (Cat. No. 554002) for 1 h at room temperature. After washing with PBST, the bands were detected by an enhanced chemiluminescence plus western blotting detection system (Amersham Biosciences). Blots were analysed by UN Scan software.

**RT-PCR and real-time quantitative RT-PCR.** After treatment, cells were washed with PBS and total RNA was extracted by using an RNeasy Mini Kit (Qiagen). RNA was quantified spectrophotometrically by absorbance at 260 nm. Total RNA (1 μg) was reverse-transcribed using the iScriptTM cDNA Synthesis Kit (Bio-Rad), according to the manufacturer's instructions. Following cDNA synthesis, PCR amplification was then performed using BIOX-ACTTM. ShortDNAPolymerase (Bioline) with primers specific for the detection of c-fos and beta-actin (Supplementary Table) were synthesised by Invitrogen. The PCR products were electrophoresed through an agarose gel and visualised by ethidium bromide staining. We performed real-time PCR using an ABI7000 instrument (Applied Biosystems) and detected the fluorescence of samples in 96-well plates with Taq Man Gene Expression Assays (Applied Biosystems), according to the manufacturer's instructions. Each 10 μl PCR reaction contained the cDNA, H$_2$O, the Master Mix (Applied Biosystems) and Probe & Primer Mix (Applied Biosystems). The mRNA levels of MCU were normalised to GAPDH. Data were analysed by using ABI7000 System Software.

**Gene reporter assay.** GFP under an NFAT promoter (gift from Dr Yuri Usachev, University of Iowa) was used as a reporter of Ca$^{2+}$-dependent gene expression. At 24–36 h after transfection with the GFP plasmid, cells were stimulated with LTC$_4$ and the % of cells expressing GFP subsequently quantified ~24 h later per field of view[35]. Cells were stimulated with LTC$_4$ in culture medium for 8 min and then medium was changed (to remove agonist) and cells were then maintained in the incubator for ~24 h prior to detection of GFP. FCCP/oligomycin was applied 5 min prior to LTC$_4$ exposure and was washed out with agonist after 8 min stimulation.

**siRNA knockdown.** siRNA against rat mitofusin 2 was from Invitrogen (stealth RNAi$^{TM}$ 5193986). siRNA against human mitofusin 2 was from Origene (Cat No: SR306670). siRNA against OPA1 was from Origene (Cat No: SR505373). siRNA against MCU was from Origene (Cat No: SR508660). siRNA against the rat mitochondrial Na$^+$–Ca$^{2+}$ exchanger was from Invitrogen

(AACGGCCACUCAACUGUCU) and human mitochondrial Na$^+$–Ca$^{2+}$ exchanger was from Origene (Cat No: SR312772). Sequences are listed in the Supplementary Table.

**Mathematical modelling.** We denote concentrations with square brackets, e.g. '[Ca$^{2+}$]', and free mitochondrial matrix or cytosolic ion concentrations with the subscripts 'mito' or 'cyt', respectively. We extended a model of NCX by Kim and Matsuoka[9] to model the accumulation/depletion of Ca$^{2+}$ within the mitochondrial matrix. The Kim and Matsuoka model predicts the net Ca$^{2+}$ flux in ions per second through a single NCX, and can be written as a single function '$f$':

$$\text{Ca\_flux} = f\left([Ca^{2+}]_{mito}, [Ca^{2+}]_{cyt}, [Na^+]_{mito}, [Na^+]_{cyt}\right). \quad (1)$$

Given a known concentration of Ca$^{2+}$ on the cytosolic side of the mitochondrial membrane ([Ca$^{2+}$]$_{cyt}$) we model the change in concentration of free mitochondrial matrix Ca$^{2+}$ ([Ca$^{2+}$]$_{mito}$) due to a given number of NCX in the mitochondrial membrane. The mitochondria are modelled as having a volume of $V_{mito} = 1.31 \times 10^{-15}$ l (a cylinder of length 1 μm and radius 0.5 μm with hemispherical caps at each end). We model the mitochondrial matrix as a 'well mixed' single compartment, so the change in total mitochondrial [Ca$^{2+}$]$_{mito\_total}$ is given by

$$\frac{d[Ca^{2+}]_{mito\_total}}{dt} = \frac{N_{NCX}\,\text{Ca\_flux}}{V_{mito}\,N_A}, \quad (2)$$

where $N_{NCX}$ is the total number of NCXes in the mitochondrial membrane, and $N_A$ is Avogadro's constant. [Ca$^{2+}$]$_{mito}$ is then given by

$$[Ca^{2+}]_{mito} = [Ca^{2+}]_{mito\_total}/\text{buffering factor}. \quad (3)$$

[Ca$^{2+}$]$_{cyt}$ is taken from our fluorescence measurements of cytosolic Ca$^{2+}$ and scaled logarithmically such that the low concentration is 100 nM and the high concentration is either 1.5 or 15 μM (as described in the main text, and shown in Fig. 3g, h). [Na$^+$]$_{mito}$ is taken to be 4.54 mM and [Na$^+$]$_{cyt}$ as 10 mM. We do not model changes in sodium concentrations as these levels are far above those for [Ca$^{2+}$]$_{mito\_total}$ and are assumed to be roughly constant. Mitochondrial membrane potential was set to −20 mV, and temperature to 298 K. The initial condition for [Ca$^{2+}$]$_{mito\_total}$ was set to steady state before beginning the simulation. The code to run the simulations shown in Fig. 3 is available as MatLab scripts upon request.

**Statistical analysis.** Results are presented as mean ± SEM. Data were compared using Student's $t$-test or by analysis of variance (ANOVA) for multiple groups. Differences were considered statistically significant at values of $p < 0.05$.

**Data availability.** Data supporting the findings of this manuscript are available from the corresponding author upon reasonable request.

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

## Acknowledgements

This work was supported by an MRC Programme Grant to A.B.P. (grant number LO1047X). G.R.M. is supported by a Sir Henry Dale Fellowship jointly funded by the Wellcome Trust and Royal Society (grant number 101222/Z/13/Z). We thank Erwin Neher (Goettingen) for helpful discussion and comments on the manuscript.

## Author contributions

K.S. performed experiments. G.R.M. and A.B.P. developed the mathematical model. K.S. and A.B.P analysed data. K.S. G.R.M. and A.B.P. designed experiments and wrote the paper.

## Additional information

**Competing interests:** The authors declare no competing financial interests.

