## [Peer Review File · Nature Communications]

Editorial Note: This manuscript has been previously reviewed at another journal that is not operating a transparent peer review scheme. This document only contains reviewer comments and rebuttal letters for versions considered at Nature Communications. Mentions of prior referee reports have been redacted.

Reviewers' comments:

Reviewer #1 (Remarks to the Author):

In this study Samanta et al study the role of radical mitochondrial de-energization on cytosolic and mitochondrial Ca oscillations. Interestingly, they find that while in control cells cytosolic and mitochondrial oscillation are abolished, they are sustained in MFN2 KO cells. They further show that KD of NCLX, use of an inhibitor or expression of a DN mutation abolish the oscillations; underscoring the role of NCLX in this process. Based on IH and IF analysis they further suggest that there is interaction between NCLX and MFN2. Based on Ca transport analysis of permeabilized cells and modeling they conclude that these oscillations are linked to NCLX acting in a switch between reverse and forward mode. Finally they claim that the reversal of NCLX has an effect on cell survival.

This is a very interesting and important paper by a leading group in Ca signaling field. The scenario that they address is a very extreme one, a complete mitochondrial de-energization that is rarely encountered even following ischemic or hypoxic insults. Their results provide a critical glimpse to a fundamental and unanswered question on how Ca influx persists following MCU KO. Several issues should be addressed:

- 1) The mitochondrial permeability pore opens upon such depolarization, and because it dramatically increases mitochondrial Ca permeability it may play a role in the oscillation and/or in providing Ca required for NCLX activation. Therefore experiments as in fig 1 d should be conducted with or without the PTT inhibitor or KD of the MPP components, i.e. cyclophilin D.
- 2) It would be nice to show, by WB, the extent of NCLX KO and NCLX DN overexpression.
- 3) In Fig 3, reversal of NCLX is of course not only of Ca but is coupled to Na, and Na is in fact the electrogenic component of the cycle. Therefore, reversal of the Na transport will provide a stronger and more convincing indication for the NCLX reversal. This could be done in permeabilized cell models, using the classical Na dyes such as Corona Green.
- 4) While the low Na nicely inhibits the mitochondrial Ca efflux and Li, serving as an NCLX substrate, rescues it. I would expect that the low Na will enhance the reversal phase; consistent with the suppression of the reversal by the ouabain pretreatment (Fig 1). A better way to do this experiment is to keep the ionomycin condition uniform and add the Li⁺/Na⁺ mix during the IP3 phase. An even better option is to leave out IP3 and simply add Ca²⁺ to the permeabilized cells. Also, the authors suggest that there is a need for a small increase in mitochondrial Ca to start a cycle of NCLX reversal/forward switch. But what is the physiological trigger of that? Presumably MCU is the activator of the process, it would therefore be nice to do the same experiment without RR. This experiment will enhance the physiological relevance of this study. It would be important also to apply the MPP blockers and see their role in the permeabilized cells as described in point 1. Also, what is the effect of the MFN2 KO on MCU dependent Ca influx in F/O cells, this may also provide an explanation for this interesting story.
- 5) I am not sure regarding data on fig 4. The cells are fully depolarized and de-energized and therefore in most criteria would be considered severely compromised or mostly dead. I believe that the mechanistic story is good enough without it.
- 6) I would suggest that addition of a tentative scheme describing the role of each player MCU, MFN2, NCLX and MPP. Such scheme will be informative and enhance the clarity of this very nice story.

Reviewer #2 (Remarks to the Author):

The manuscript by Samanta et al describes the role of the mitochondrial sodium-calcium exchanger (NCX) in shaping mitochondrial calcium oscillations. The authors show that the NCX switches between forward and reverse mode during calcium oscillations. This demonstrates that the same molecule can be responsible for calcium entry and release. Overall, the manuscript is extremely well written and addresses an important point in calcium signalling. The data provide compelling evidence for the tested hypothesis, and I particularly like the link between mitochondrial calcium oscillations and gene expression patterns. Once the minor comments listed below have been addressed, the manuscript can be accepted for publication.

- Comparing Figures 1C and 1G, it seems that there are fewer peaks in the mitochondria than in the cytosol. Could the authors comment on whether this is an issue with the data analysis, or what mechanism would lead to the discrepancy in the number of peaks?

- The authors identified mitofusin 2 as a key component in regulating mitochondrial calcium oscillations. Using pull downs, they show that mitofusin 2 interacts, directly or indirectly, with the mitochondrial NCX, but do not provide a further discussion of this point. Could the authors expand on the potential direct or indirect interactions?

- Figure 3I does not seem to add much to the manuscript. It basically shows that increasing the number of NCXs increases peak mitochondrial calcium, and that increasing buffering decreases it.

Reviewer #3 (Remarks to the Author):

In this manuscript, Samanta et al. explore a fascinating connection between mitochondrial Na⁺-Ca²⁺-exchanger (NCLX) and mitofusin (MFN2), and the role of these proteins in mitochondrial/cytosolic Ca²⁺ signaling. They demonstrate that knockdown of MFN2 rescues oscillatory cytosolic and matrix Ca²⁺ signals in cells under the conditions where mitochondria are de-energized (with the use of FCCP and oligomycin). They further demonstrate that NCLX is essential in this process and that it involves sequential switching between forward and reverse mode of Na⁺-Ca²⁺-exchange. They propose that NCLX mode switching is regulated by MFN2. Finally they show that functional NCLX supports gene expression and delays apoptosis in cells with de-energized mitochondria. Although majority of the experiments are performed in RBL-1 mast cells, the conclusions are likely to be applicable to other cell types.

The experiments are generally well thought out, and well executed. My only objection concerns the co-immunoprecipitation in Fig. 2I, which is presented as evidence for direct interaction between NCLX and MFN2. In fact, the ratio of MFN2 to NCLX in the precipitate is exceedingly low, far lower than in the extract. Although it is not unusual to see poor preservation of protein complexes under Co-IP conditions, the experiment shown is totally unconvincing. Furthermore, a proper control would require an unrelated IMM localized myc-tagged protein (not IgG as presented). Since the issue of direct NCLX-MFN2 interaction is marginal in the context of this manuscript, I would suggest to delete panels G,H and I in Fig. 2, and the relevant text in the manuscript.

The manuscript could be improved by including a brief discussion of the role of mitofusin in mitochondrial fusion dynamics and mitochondria-ER/SR connection, especially in the context of heart muscle contractility, which was the subject of several recent publications.

Reviewer #4 (Remarks to the Author):

The present study manuscript by Krishna Samanta and coworkers describes the maintenance of

cytosolic and mitochondrial Ca oscillations after mitochondrial membrane potential ($\Delta\psi$) collapse when the mitochondrial fusion protein mitofusin2 is silenced in the RBL1 mast cell line. They provide reasonable evidence that that these oscillations originate from the activity of the mitochondrial Na/Ca exchanger (NCLX) cycling between forward and reverse modes. While I have only a few problems with the experiments presented, my major concern is that the study is overly descriptive, provides few mechanistic insights and most importantly reports a phenomenon with no connection to physiology or pathophysiology.

First, the conditions described in this study with membrane potential being completely collapsed with protonophore, in combination with inhibition of ATP synthesis, together with the knockdown of mitofusin, will never occur under physiological conditions and thus the observations are purely phenomenological and thus can not be plausibly related to any cell function. Specifically, the activity of the Na/Ca exchanger is absolutely predictable based on the electromotive driving forces which are artificially imposed over the mitochondrial membrane. Under these conditions, it is not surprising that any cytosolic signal would propagate to these mitochondria. Furthermore, this "reverse mode" of Na/Ca exchange in mitochondria has been reported previously under similar conditions (Kim and Matsuoka. *J. Physiol.* 586 1683-97 (2008) and is thus, not even a novel observation.

The one potentially interesting aspect of this manuscript is the rescue of cytosolic Ca signals and subsequently mitochondrial signals by knockdown of mitofusin2, apparently utilizing a fusion-independent mechanism. What is quite remarkable is that MFN2 KD seems to reverse the effects on mitochondrial potential collapse without affecting membrane potential, which begs the obvious and interesting question of how MFN2 KD achieves this? The study however, provides no real mechanistic clues on the action of mitofusin2 within this context, beyond the fact that the 2 tagged, overexpressed proteins appear to co-IP and their distribution overlaps within the cell. The experiments are wholly inadequate to discern mechanism and even experimentally they are superficial; there is no mention of how many times that the experiments were performed and no quantification. In addition, it is surprising that 2 presumably, appropriately targeted, overexpressed mitochondrial proteins do not overlap in distribution in cells at rest (2G) at the resolution of confocal microscopy.

Most experiments make use of protocols that are extremely difficult if not impossible to interpret where a double knockdown (MFN2 and NCLX or MFN2 and MCU) plus a combination of 2-4 drugs (FCCP, oligomycin, CGP, Ionomycin). See e.g. page 4 "...In cells in which we reduced both MCU and mitofusin2 expression and then depolarized mitochondria, oscillations in matrix Ca to leukotriene receptor stimulation were rescued only if matrix Ca had been increased slightly by Ionomycin prior to agonist exposure". The assertion by the authors that "Collectively, these results show that FCCP-induced Cell death is delayed by functional mitochondrial Na/Ca exchange" is not accurate; cell death would be delayed by MFN2 KD instead for which the mechanism is unresolved.

Other issues

The authors provide no data to confirm that NCLX knockdown is effective.

FCCP will collapse H⁺ gradients across all membranes. How do the Authors exclude the fact that cytosolic pH changes may contribute to this phenomenon.

The suggestion by the authors that their findings could explain why MCU KO mice show an unexpectedly mild phenotype is misleading. MCU KO mice are viable only on a mixed background and their mitochondrial membrane potential and matrix Ca²⁺ are not significantly affected. Under these homeostatic conditions of hyperpolarized mitochondrial membrane potential, it is biophysically impossible for NCLX to work in reverse mode and underlie an alternative Ca import pathway.

Reviewers' comments:

Reviewer #1 (Remarks to the Author):

Grundmann et al., investigate the independence and interdependence of G-protein and β -arrestin signaling downstream of GPCRs using a combination of “clean” KO background, inhibitors, receptor mutants and ligand variants. Unfortunately, the majority of conclusions drawn in this study do not provide any novel information, and in a very few places where they do, they refute the large body of data currently available from multiple laboratories.

Response authors: we thank the reviewer for his comments. Clearly, our original manuscript lacked sufficient clarity to unambiguously convey the novelty of our findings. In our revised manuscript, we paid particular attention to enhance the clarity of presentation and to highlight the novelty of our findings. Clearly, our findings are surprising and novel and should be of great interest for a broad readership. In agreement with this notion, one of the other referees considers our study as 'paradigm shifting'. This is what we feel as well. In addition, the quality of the manuscript and the strength of our conclusions have been further enhanced by incorporating a series of new data into the revised manuscript, as requested by the other reviewers.

As discussed in detail in the manuscript, our findings are novel and paradigm-shifting. Our perception of how GPCRs signal have taken intriguing twists over the last decades. I cite from a review of Robert Lefkowitz (Curr Opin Cell Biol 2014): ... it was quite surprising when it was first noted in the mid 90's that, at a single GPCR, different ligands could be biased or functionally selective toward one or another of these G proteins. Even more surprising were the discoveries a few years later that GPCRs could also signal through beta-arrestins and that **ligands could be biased towards either a G protein or beta-arrestin-mediated pathways ...**”

Beta-arrestin-mediated activation of ERK is one of the most prominent examples of arrestin-dependent, G protein-independent signaling. Yet, our work with G protein- and arrestin-depleted cells indicates that it is the other way round: **G proteins drive ERK activation and arrestins do not, a clear paradigm shift.**

Major points:

i) For example, the lack of β arr dependent ERK activation for β 2AR has been published recently (Science Signaling, 2017). Authors do not properly cite and discuss this thorough paper which basically reaches very similar conclusions at least for the β 2AR.

Response authors: we apologize for not discussing this work in more detail. The study by the Gutkind lab reaches similar conclusions for the β 2 receptor. However, our study greatly extends the work by Gutkind et al. by investigating a broad panel of GPCRs and using different experimental approaches including DREADD technology. It is also important to note that our work was carried out completely independently of the Gutkind study. Thus, the Science Signaling paper and the present study very nicely complement each other by providing converging novel insight into the role of beta-arrestins in GPCR signaling.

We now refer to Science Signaling study in more detail in the revised version of the manuscript (please see Introduction, page 3, last sentence, and Discussion, first paragraph, yellow text).

ii) Using CRISPR based approach to generate “clean” background devoid of G proteins or β arrestins to investigate GPCR signaling have been described previously (JBC, 2017; Science Signaling, 2017 and a few others).

Response authors: we are aware of several papers in which individual $G\alpha$ proteins or beta arrestins have been eliminated by CRISPR/Cas9 genome editing and cite these papers in our manuscript (e.g. refs 22,23,24,28,67). We were also very careful **not to claim any credit** for generating these **published** CRISPR/Cas9 HEK cell lines. **However, we do claim novelty for establishing cellular conditions that have not been created before: “zero functional G”**, which stands for the collective absence of functional $G\alpha$ proteins from all four major G protein families. Arrestin signaling in general, or more precisely, how GPCRs activate ERK, has never been studied under these conditions. To emphasize the novelty of our findings, we also chose a new title for our manuscript: **“Lack of beta-arrestin signaling at ‘zero functional G’”**.

iii) There is pretty much only lab (or collaborators thereof) in the community which uses DMR based approach, and honestly, it is difficult to convincingly state what exactly it reports in cellular context.

Response authors: Indeed, our lab pioneered the application of DMR to analysis of GPCR signal transduction (Schröder et al., **Nature Biotechnology 2010**; Schröder et al., **Nature Protocols, 2011**). However, a current literature search indicates that DMR technology is now widely used in the GPCR community. Please see specific examples of papers below that were published **without contribution of our group**:

¹ Biophys J. 2006 Sep 1;91(5):1925-40. Epub 2006 Jun 9. Resonant waveguide grating biosensor for living cell sensing. Fang Y, Ferrie AM, Fontaine NH, Mauro J, Balakrishnan J.

² Nat Chem Biol. 2011 Oct 23;7(12):909-15. doi: 10.1038/nchembio.690. GPCRs regulate the assembly of a multienzyme complex for purine biosynthesis. Verrier F, An S, Ferrie AM, Sun H, Kyoung M, Deng H, Fang Y, Benkovic SJ.

³ FEBS Lett. 2013 Aug 2;587(15):2399-404. doi: 10.1016/j.febslet.2013.05.067. Epub 2013 Jun 13. Succinate receptor GPR91, a $G\alpha(i)$ coupled receptor that increases intracellular calcium concentrations through PLC β . Sundström L, Greasley PJ, Engberg S, Wallander M, Ryberg E.

⁴ J Pharmacol Toxicol Methods. 2017 Nov;88(Pt 1):72-78. doi: 10.1016/j.vascn.2017.07.003. Epub 2017 Jul 15. Applying label-free dynamic mass redistribution assay for studying endogenous FPR1 receptor signalling in human neutrophils. Christensen HB, Gloriam DE, Pedersen DS, Cowland JB, Borregaard N, Bräuner-Osborne H.

⁵ Mol Pharmacol. 2012 May;81(5):631-42. doi: 10.1124/mol.111.077388. Epub 2012 Jan 26. Differential signaling by splice variants of the human free fatty acid receptor GPR120. Watson SJ, Brown AJ, Holliday ND.

⁶ Pharmacol Res. 2016 Mar;105:13-21. doi: 10.1016/j.phrs.2016.01.003. Epub 2016 Jan 7. Dynamic mass redistribution reveals diverging importance of PDZ-ligands for G protein-coupled receptor pharmacodynamics. Camp ND, Lee KS, Cherry A, Wacker-Mhyre JL, Kountz TS, Park JM, Harris DA, Estrada M, Stewart A, Stella N, Wolf-Yadlin A, Hague C.

⁷ Pharmacol Res Perspect. 2014 Feb;2(1). doi: 10.1002/prp2.24. Dynamic mass redistribution analysis of endogenous β -adrenergic receptor signaling in neonatal rat cardiac fibroblasts. Carter RL, Grisanti LA, Yu JE, Repas AA, Woodall M, Ibeti J, Koch WJ, Jacobson MA, Tilley DG.

⁸ Pharmacol Res. 2016 Dec;114:13-26. doi: 10.1016/j.phrs.2016.10.010. Epub 2016 Oct 15. Label-free versus conventional cellular assays: Functional investigations on the human histamine H1 receptor. Lieb S, Littmann T, Plank N, Felixberger J, Tanaka M, Schäfer T, Krief S, Elz S, Friedland K, Bernhardt G, Wegener J, Ozawa T, Buschauer A.

⁹ Pharmacol Res. 2016 Jun;108:39-45. doi: 10.1016/j.phrs.2016.04.018. Epub 2016 Apr 23. Label-free cell phenotypic profiling and pathway deconvolution of neurotensin receptor-1. Hou T, Shi L, Wang J, Wei L, Qu L, Zhang X, Liang X.

¹⁰ Cell Signal. 2015 Apr;27(4):818-27. doi: 10.1016/j.cellsig.2015.01.008. Epub 2015 Jan 22. Identification of amino acids that are selectively involved in Gi/o activation by rat melanin concentrating hormone receptor 1. Hamamoto A, Kobayashi Y, Saito Y.

¹¹ J Pharmacol Exp Ther. 2015 Mar;352(3):480-93. doi: 10.1124/jpet.114.220293. Epub 2014 Dec 24. Bias analyses of preclinical and clinical D2 dopamine ligands: studies with immediate and complex signaling pathways. Brust TF, Hayes MP, Roman DL, Burris KD, Watts VJ.

¹² Assay Drug Dev Technol. 2014 Aug;12(6):361-8. doi: 10.1089/adt.2014.590. A comparison of assay performance between the calcium mobilization and the dynamic mass redistribution technologies for the human urotensin receptor. Lee MY, Mun J, Lee JH, Lee S, Lee BH, Oh KS.

Regarding the question as to what DMR reports in cellular context: DMR is a phenotypic method providing a real-time integrated signal of a cellular response that originates from morphology changes of cells that are grown on biosensor plates. We explain this method briefly in the beginning of the DMR data section and refer to literature describing the biophysical and cellular basis of DMR detection (Biophys J. 2006 Sep 1;91(5):1925-40. Resonant waveguide grating biosensor for living cell sensing; Fang Y et al.).

iv) There are multiple publications from different laboratories establishing the arrestin-biased nature of SII at AT1R; however, the authors cite the single paper which claims that SII can induce some G protein coupling (reference 40). It is fine to discuss this if it supports their finding but a balanced discussion should be included.

Response authors: we thank the reviewer for this comment. We fully agree with the arrestin-biased nature of SII, meaning that it clearly **prefers arrestins over G proteins**. SII was introduced into the literature as **completely arrestin-biased** by Bob Lefkowitz and coworkers (Wei et al., PNAS 2003). About a decade later, the **very same lab** provided a **more accurate description** of the 'arrestin-biased' nature of SII and refined the molecular mechanism of SII action to reveal its **relative preference for arrestins over G proteins** rather than an absolute bias for arrestins (Strachan et al., JBC 2014). We feel that this seminal publication is highly relevant for the field because it helped to clarify the mode of SII action. Therefore we cited this work in addition to the previous ref 40. We also rephrased the Discussion to emphasize that a thorough mechanistic understanding of how AT1R activates ERK requires consideration of the full spectrum of biological activities of SII, as revealed by the seminal work of the Lefkowitz group (Strachan et al., JBC2014) (page 11, third paragraph).

v) Lines 330-339 – authors write that there is no study on tail vs. core conformation of GPCR-bound β arrs. This is not correct – there are three papers (Nature Communications, 2016; PNAS, 2017; Mol Biol Cell, 2017) that clearly establish the sufficiency of tail conformation to support receptor endocytosis and ERK activation.

Response authors: it was recommended by another referee to remove the above paragraph from the discussion due to its overly speculative nature. Still, we thank this reviewer for pointing out these recent elegant studies, two of which are also cited in our revised manuscript (ref 22,23).

vi) Control of surface population of GPCRs at “ZERO G” by β arrs is not only intuitive (owing to constitutive endocytosis) but also documented experimentally (PNAS, 2016).

Response authors: we thank the reviewer for this comment and assume that he/she refers to recent work of the Benovic lab (Carr et al., PNAS 2016), among other studies (the Carr et al. paper has been included in our reference list). Carr et al. utilized an arrestin-biased ligand to demonstrate arrestin-biased signaling to the ERK cascade. However, this study differs from our work in that it is **not done at ‘zero functional G’**. Instead, an arrestin-biased ligand was employed in cells with functional G proteins. While we understand that this may be interpreted as zero G from the ligand’s perspective, our study is at **‘zero functional G’ from the cell’s perspective** for which there is **no precedence to date**. Thus, the tools employed in both studies are vastly different, as are the resulting mechanistic conclusions.

Similarly novel and paradigm-shifting was a recent report from Gosh and Shukla (Nature Nanotechnology 2017). The authors developed an intrabody as generic inhibitor of GPCR endocytosis to realize that indeed, endocytosis and ERK signaling, can be separated for all receptors, contrary to the long-held view that internalization is necessary for β arr-driven ERK signaling.

vii) Interdependence of ERK signaling, at least for G_i coupled receptors, has been documented by many studies, and therefore, the requirement for some level of G protein for β arr dependent signaling is not surprising.

Response authors: we very much agree with this reviewer that the requirement for some level of G protein for β arr dependent signaling is not surprising. What is highly surprising is that – despite of this knowledge – G protein-independent β arr-dependent signaling is widely postulated, even for G_i -coupled receptors, despite the fact that G_i -mediated ERK phosphorylation is entirely sensitive to PTX. For example, Walters et al. (JCI 2009) stated in the abstract “In a human cell line-based signaling assay, nicotinic acid stimulation led to pertussis toxin-sensitive lowering of cAMP, recruitment of beta-arrestins to the cell membrane, an activating conformational change in beta-arrestin, and beta-arrestin-dependent signaling to ERK MAPK”. This implies that cAMP formation is driven by G_i but ERK-MAPK signaling is not, although the results section shows complete dependence of ERK on PTX pretreatment. These statements suggest that arrestins drive ERK and that G proteins are dispensable. Yet, our work with G protein and arrestin-depleted cells indicates that it is the other way round: **G proteins drive ERK activation and arrestins do not, a clear paradigm shift.**

viii) Authors mention that incomplete suppression of β arrs in HEK cells in earlier studies might explain the discrepancies observed here. However, many of the studies have used KO MEFs which

has as “clean” a background as CRISPR-CAS approach. At least for β 2AR, Gs-null cells have also been used to document β arr dependent ERK activation (JBC, 2004).

Response authors: we thank the reviewer for raising this point. Mouse embryonic fibroblasts have been used in several studies to understand how beta receptors activate ERK. However, these studies have not led to a consensus view. Shenoy and Perkovska et al. (refs 7,62) claim **G protein-independent, arrestin-dependent** ERK activation; Sun et al. (ref 28) propose **arrestin-independence for this response**; Huang et al. (ref 60) show **Gs-dependence but arrestin-independence**, and Coffa et al. (ref 61) propose **arrestin-dependence**. We refer to these studies in the Discussion section and indicate that results obtained with MEFs have not provided a consensus regarding the mechanisms of how GPCRs drive ERK in this cellular background.

Reviewer #2 (Remarks to the Author):

Here the authors addressed one of the hottest topics in the GPCR field: the issue of G protein-independent b-arrestin-dependent signaling. The authors should be commended for several aspects of their study. First, by using genetic elimination of Ga subunits of particular families and of both b-arrestins with CRISPR-Cas, they avoided ambiguities of siRNA or shRNA knockdown, which is virtually never complete and sometimes results in the knockdown of non-targeted proteins, misleading and undermining interpretation (e.g., compare EMBO Rep 2010, 11 (8) 605-11 and EMBO Rep 2013, 14 (2), 164-71). Second, they tested a large number and variety of GPCRs, including prostaglandin D2 receptor DP2, orphan receptor with synthetic agonist GPR17, free fatty acid receptor FFA2, as well as model receptors used in studies where b-arrestin-dependent signaling was first described, b2-adrenergic receptor, vasopressin V2 receptor, and angiotensin AT1 receptor. Finally, the study was complemented with unbiased, Gq-biased, and b-arrestin-biased M3-based DREADDs. Third, wherever possible, they used ligands of these receptors that were reported to be biased. Fourth, in addition to commonly used readouts (cAMP, IP accumulation, ERK phosphorylation), they also used global mass redistribution (DMR) readout, which is not dependent on an individual pathway. Fifth, in addition to signaling readouts, they tested direct b-arrestin recruitment and b-arrestin-dependent GPCR internalization. Thus, in terms of comprehensiveness, this study has few peers. Finally, the authors present all necessary controls.

The results suggest that in all cases detectable signaling requires the activation of G proteins, whereas arrestin recruitment and receptor internalization does not. In some cases the presence of b-arrestins enhances ERK1/2 phosphorylation, suggesting scaffolding functions of these proteins downstream of G proteins. Thus, the authors conclude that there is b-arrestin-dependent GPCR signaling, but it is not G protein-independent.

The data presented are a lot less ambiguous than the results of most previous studies, where protein knockdown was employed, and therefore much more convincing. The authors logically explain previous reports in the context of their model. Overall, this work appears to be paradigm shifting. It would be of great interest to numerous researchers in the fields of GPCRs, cell signaling, and drug discovery.

However, several minor changes can further improve the manuscript:

Response authors: we thank this reviewer for the very favorable evaluation and constructive comments on our manuscript, and for regarding this work as **paradigm-shifting**. Thus, **contrary** to the **long-held view** that **arrestins are signal transducers in their own right, they are not, at least not under the experimental conditions used in our study**.

1. The data in Suppl Fig 8 are also consistent with a known fact that simple scaffolds (i.e., those that bring signaling proteins together but do not activate either of them) demonstrate bell-shaped dependence of signaling on scaffold concentrations (theory: PNAS 2000, 97 (11), 5818-23; experimental evidence: Biochemistry 2011, 50 (48), 10520-9; JBC 2013, 288 (52), 37332-42). At relatively low scaffold levels the signaling is enhanced via the formation of complete complexes scaffold-molecules A+B, whereas when the number of scaffold molecules exceeds the number of proteins they scaffold, the signaling is suppressed via preferential formation of incomplete complexes (scaffold-molecule A and scaffold-molecule B, but not scaffold-molecules A+B).

Response authors: we thank the reviewer for providing an alternative explanation to our data. In fact, we do prefer this reviewer's explanation over our own. Therefore both rationalization of our experimental results, as suggested by this reviewer, and the associated literature are now part of our revised manuscript.

2. References where modified M3-based DREADDs were characterized are necessary.

Response authors: we apologize for this omission. Three references are now given for the wildtype, Gq-biased, and arrestin-biased form of the M3-DREADD (refs 51,52,53).

3. Some minor editing is needed: lines 198 and 292, by "barr recruitment" the authors clearly mean "barr-dependent signaling"; line 337, the word "pendant" is inappropriate here and should be replaced.

Response authors: as suggested, we replaced "barr recruitment" by "barr-dependent signaling" or "barr signaling" (page 5, bottom, yellow highlighted text and page 7, bottom, yellow highlighted text). The term "pendant" was removed by deletion of a paragraph (previous lines 330-339) judged as 'overly speculative' by another referee.

Reviewer #3 (Remarks to the Author):

This paper deals with the question of involvement of G protein in arrestin-dependent signalling – particularly to pERK MAPK.

While I believe that the work addresses an important aspect of GPCR signalling and is of potential value to the field, there are multiple issues that need to be addressed.

While it is true that there are groups that talk about "G protein-independent" arrestin dependent signalling (and this tends to be motivated by the focus of these groups on arrestin-dependent signalling), this is not the consensus view for the field. I would argue that most competent researchers in the field have an expectation of an involvement of G protein (and in particular the $\beta\gamma$ -subunit) in regulation of arrestin recruitment/activation/signalling. Most groups would describe this

signalling as simply “arrestin-dependent” rather than also G protein-independent. As such the introduction should be toned down to more appropriately reflect the broad spectrum of views in the field, though it is appropriate to acknowledge that G protein-independence is claimed by some groups.

Nonetheless, the authors are correct in that how these interplay is an area that has not been studied with rigour, and as such, the current paper is potentially important in helping to define this.

For this reviewer, the results are not surprising, but direct, robust experimental evidence is not readily available in the literature, and there are groups that inappropriately ascribe G protein-independence to particular signalling, without the required evidence for this statement. Studies such as those reported in the current paper are therefore important to address this, but equally they need to be cautious in their description of their data, and additional work is required to support the current claims.

Response authors: we thank this reviewer for recognizing the importance of our manuscript and for her/his constructive comments.

Comments

Intro – while the DMR label free experiments measure integrated responses, it is not correct to term this “pathway-unbiased”.

Response authors: As requested by the reviewer, we **deleted** the term "**pathway-unbiased**" in the revised manuscript version.

Line 61, “how they interfere” – do you mean “how they interplay”?

Response authors: “interfere” has been replaced by “interplay”

Lines 65-66. Signaling should not be described as “unbiased”, “G protein-biased” etc, as absolutes. These behaviours are always contextual, dependent upon cell background, require context of pathways studied (and normally the comparator ligands), and require reference as to what the “relative bias” actually describes.

Response authors: we thank the reviewer for raising this critical point. It is very common that ligands and receptors are simply referred to as ‘balanced’ or ‘biased’, although it is known that bias is strongly affected by various factors including cellular context. To address this issue, we made several changes throughout our manuscript:

- we have deleted the term from the abstract and no longer refer to ‘unbiased’ and ‘arrestin-biased ligands’
- we placed the terms ‘biased’, ‘unbiased’ within quotation marks throughout the manuscript (this is frequently done by authors who are aware of the conditional nature of this term (see for example lit. by Kenakin (ref 66) or Cheloha et al. (ref 67))).
- we explicitly mention the cellular context in which the ‘unbiased’, ‘Gq-biased’ and ‘arrestin-biased’ nature of M3-DREADDS was defined (page 8, lines 1-3, highlighted in yellow).

- we have rephrased the Discussion regarding the nature of SII bias to indicate that the authors are aware of the conditional nature of this term (page 11, third paragraph, text highlighted yellow).

Lines 67-68 – “true zero G” – I don’t think this is an appropriate term to use, based on the current experiments. I am ok if they wish to use “zero functional G”, at least in the context of a-subunit dependent function. However, unless the authors demonstrate that under conditions of over-night PTX treatment there is no G protein present, or minimally that the PTX treated Gi proteins are unable to be recruited to the receptors, then calling the condition “zero G” is misleading. For example, dominant –ve mutant Gi proteins are effectively recruited to activated receptors. This term should not be used.

Response authors: we fully agree with the reviewer. The term "zero functional G" is clearly more appropriate to describe the experimental conditions we have chosen. We replace the term ‘zero G’ with ‘zero functional G’ throughout the entire manuscript.

Line 73 – the end of this sentence should provide the caveat that G protein-independence has not been robustly demonstrated (which is the point of the current paper).

Response authors: we now raise the caveat that arrestin-dependence but not G protein-independence has been extensively studied (page 3, last sentence of Introduction, highlighted yellow).

Lines 78-80 – I would suggest noting that the G protein partners identified for the studied receptors are “canonically studied partners”, unless they have evidence that no other G protein coupling occurs (in any system).

Response authors: done, we added ‘canonically studied signaling partners’ (page 4, line 5)

Results para 1. As noted above – the authors do not demonstrate “G protein-independent” recruitment of arrestin, only recruitment in the presence of functional disruption of the Gai subunit. The use of the term “Barr recruitment at zero G” should be revised to reflect the actual experimental evidence.

Response authors: As stated above, we have replaced ‘zero G’ with the term ‘zero functional G’ which is much more appropriate. In addition, we provide new experimental evidence with genetically encoded **FRET-based biosensors that G proteins are not functional when cells are inhibitor-treated**. This new set of data further **strengthens the term ‘zero functional G’**, as we demonstrated absence of G protein functionality using canonical second messenger assays (already present in the previous version), and, additionally, absence of activation-induced conformational changes within G protein heterotrimers (**new data in Figure 1, panels b,f,g,l,m**).

Results – general comment

With the exception of a limited subset of DMR experiments and cAMP, IP1 assays, there is no concentration-response data. This is true for b-arr recruitment (with the exception of Fig 1 B- side note – all figures showing BRET for b-arr should note that it is for arrestin on the Y-axis), and there is no concentration-response data for pERK. These latter 2 assays are the most critical for the

arguments that are developed and require proper quantification (not just single high-concentration drug). The DMR may be easier (higher through-put) for the authors, but it only provides confirmatory data (as it measures a highly integrated response). All BRET and all pERK assays should be done as concentration-response.

Response authors: we thank the reviewer for this comment. We have **replaced ALL BRET** assays in the main body manuscript with **concentration-effect data (Figure 1, panels h,i,n,o)**. We have performed **all pERK assays of Figure 3** in concentration-response mode (**Supplementary Figure 4d,e,f**). Concerning proper quantification of pERK data: we chose to analyze **one single high concentration on purpose** to quantify pERK at various time points rather than cover a broad range of concentrations at a single time point. Our study was designed to interrogate the role of arrestins as signal transduction drivers using phenotypic integrated cell responses (DMR) and pERK assays. **Independent G protein and arrestin pathways** have been postulated to mediate angiotensin II AT1R-receptor-dependent ERK activation (Wei et al., PNAS2003; Ahn et al., JBC2004), with **G proteins** responsible for the **rapid early** and **arrestins** for the **delayed component**. With the goal of **discriminating arrestins versus G proteins as ERK drivers**, we considered it important to monitor responses over a **wide time-frame**. Therefore we chose a **single high ligand concentration** but **varied the parameter time**. We felt that this is the most sensible approach for identifying ERK drivers.

We **did perform pERK assays for all receptors shown in Fig3 in concentration-effect mode, as requested by the reviewer**. We arrived at the **same conclusion obtained with the single high concentration-effect data**: G proteins but not arrestins drive ERK signaling in their own right. To **focus on driver** but not scaffolding functions of arrestins, we performed **ALL assays in kinetic mode**.

Please see the below publication to name but one in support in of our experimental strategy:

“Kumari et al., Nature Communications 2016: “Of particular interest is the ERK activation at late time points (10, 20, 30 min) which are well established to be mediated by barr-dependent and G protein-independent pathway” (ref 23 in our manuscript).

Lines 150-154 – The text with respect to EGF-dependent stimulation of pERK is misleading, as it implies that this is not altered, but a comparison of measures on the Y-axes illustrates that this signalling is also attenuated in various treatment conditions (including b-arr^{-/-} cells).

Response authors: we thank the reviewer for pointing out this misleading statement. In fact, EGF is very suitable as viability control to ascertain that inhibitor treatment is specific, yet it is not suited as reference for quantification of receptor stimuli or comparison of signal strength for receptor stimuli across different cell lines. In other words: larger EGF signals do not always coincide with larger receptor signals and EGF signals vary also with the nature of receptor expressed in a given cellular background. For these reasons, EGF is **only used as viability control** for inhibitor treatment and in cells in which receptor stimuli are undetectable, i.e. G protein-depleted cells.

Appropriate changes have been made in the revised manuscript (page 6, lines 11-13).

Lines 161-162 – as noted above, though it is correct that the described signalling has been described by some groups as “G protein-independent” arrestin-dependent – this is not universally true. This

description promotes an impression of absolute opinion within the field. A more balanced statement would be appropriate.

Response authors: we thank the reviewer for raising this point. We realize that our chosen terminology may even imply that the three family A receptor prototypes exclusively signal in a G protein-independent, arrestin-dependent fashion. This is certainly not the case and does not reflect our intended meaning. We rephrased the misleading statement by a more balanced one (page 6, last paragraph, lines 1-5, altered wording highlighted yellow).

Line 164, isoprenaline cannot be described as a full “unbiased” agonist. This is not a sustainable statement as an absolute (see comments above). Similarly, carvedilol, although described by some authors as G protein-independent (line 165), this should not be presented as a universal description.

Response authors: isoprenaline is now referred to as ‘synthetic full agonist’. Regarding carvedilol: we now state that only **some** groups refer to it as partial agonist for β -arrestin-mediated, G protein-independent ERK1/2 signaling (page 6, last paragraph, last three lines).

Line 171, Ang II cannot be described as an “unbiased” agonist without detailed contextual statements that define the specific conditions under which this was determined.

Response authors: Ang II is now referred to as natural agonist rather than as ‘unbiased’ ligand and the cellular context in which the arrestin-biased nature of SII has been identified is now indicated (page 7, lines 5,6).

Line 204, as above “unbiased” M3D-WT – can’t use this term without context.

Response authors: ‘unbiased’ as well as ‘Gq-biased’ and ‘arrestin-biased’ DREADDs are placed within quotation marks to indicate that bias is not absolute, and cellular context is also given, along with three citations referring to generation and characterization of these DREADDs in the COS7 and HEK293 cell background (refs 51,52,53).

Line 205, DREADDs are poorly response to ACh rather than “unresponsive”

Response authors: as suggested, unresponsive was replaced with poorly responsive (page 8, lines 2,3)

Lines 213-214, “all cells without Ga retained EGF responsiveness” – this is true, but they need to note that it is also reduced relative to “WT”.

Response authors: done, we added reduced pERK for EGF in relation to responses observed in WT cells (page 8, lines 11,12). As already stated above, we used EGF as viability control but not as reference ligand for normalization of receptor-mediated pERK responses. This is due to the lack of a strict correlation between EGF and GPCR signal strength.

Lines 218-219, also a general comment on analyses. Firstly, statistics – these are not described in methods, and regularly there is no description of what, specifically, is compared. Regularly, there is no error on WT, basal, control etc where this has been normalised. As such, if the statistical analyses is done on data shown in graphs, it is completely invalid as it removes the error within the control.

Response authors: thank you for making us aware of the necessity to correct statistics and to enhance clarity of presentation as to what specifically has been compared. In need of statistical advice and guidance for the revision we received guidance by Dr. John Spouge (NIH statistician; spouge@ncbi.nlm.nih.gov; available for any request on revised statistics).

With this revision, we provide the requested description of statistics in the Materials & Methods. We have furthermore performed numerous corrections/changes as specified below in detail:

- In all figure legends, we describe what specifically is compared (Fig.1, Fig. S4, Fig. S6, Fig. S9).
- Supplementary Figures S3,S4,S9 were deleted (corresponding manuscript text does not claim significance) and statistics are not necessary.
- Invalid or unnecessary statistics have been removed from all datasets where the manuscript text does not make claims on significance, and statistical tests have been retained exclusively for those datasets where significance is not obvious to the eye or is needed to support specific claims (Fig.1, Fig. S4, Fig. S6, Fig. S9).
- Please note that we revised all statistics entirely based on advice given by Dr. Spouge.

In regard to comments in relation to higher expression of M3D-Gq. The data do not appear to support this. Eg. Fig S10 (no difference – even with no WT error). Fig. S12 reports a significant difference (no error for WT), and this also contrasts with data in Fig. S13 that shows raw data for expression and that it is not altered (lower in the M3D-Gq v WT if anything; panel C v panel D).

Response authors: thank you for noting these inconsistencies in the above Supplementary Figures related to surface abundance of M3-DREADDS.

- Fig.S10 (now Fig.S7, panel A) does not support higher expression of M3D-Gq in WT cells and we do not state this anywhere in the manuscript.

- Fig.S12 (now Fig. S9) reports a significant difference for M3D-Gq in the barrKO background which we explicitly mention in the text, so yes, this is correct.

- Fig.S13 (deleted in the revised version) showed raw data contradicting the previous Fig.S12, and we apologize for this mistake. The data in the previous Fig.S12 (now Fig. S9) are correct. Inspection of raw data for M3D-Gq expression show that it is enhanced in barrKO cells relative to WT cells. However, removal of a quantitative statement in the revised manuscript (comparison of pERK folds for M3D and M3D-Gq in wt and barr KO cells) makes the previous Suppl. Fig. 13 dispensable.

Lines 218-227, the commentary related to reduced pERK in delta Barr1/2 ignores the fact that the EGF response is also attenuated.

Response authors: yes, this is correct. We do not consider EGF as appropriate control for quantification of signaling strength of GPCRs. Variability within and across cell lines and lack of consistent correlation of EGF and GPCR-mediated signals are the reason for this. Therefore we removed the statement made in the original manuscript (lines 218-227) which actually distracts from the main message of the paper: **to identify drivers for pERK** rather than comment on evidence supporting a scaffolding function for barrs which, in the view of these authors, is undisputed.

Lines 224-225 "... that Barr recruitment at zero G was a salient feature of M3D-Barr". It was not clear to me how the pERK and DMR assays demonstrated this... as Barr recruitment was not measured.

Response authors: Please note: Arrestin recruitment assays have been performed for M3D-barr in WT background and also for M3D-barr and M3D-wt in the DeltaGsix + PTX cells (Figures Fig. S7B and S8B).

Para (lines 230-250) – the description implies that there is no impact on EGF signalling, however, there is considerable variation to this response (often in-line with other patterns of attenuation). Has this been evaluated statistically? Is this variance taken into consideration when comparing GPCR-dependent responses?

Response authors: we see the reviewer's point. The description states that variability of EGF responses in non-transfected clones is rather small (Fig. 6U-X), which was the case. However, upon enrichment of cells with different receptors, variability increases, and, importantly, does not correlate with the strength of receptor-dependent signals. For example, please compare panel **k** with **l**. We chose to share these data but did not use EGF for normalization of receptor stimuli, because there is no obvious correlation between EGF and GPCR-dependent signal amplitudes.

Lines 254-255, this should include the caveat "in this cellular background"

Response authors: done, "in this cellular background" was added to the sentence as suggested (page 9, line 16, yellow highlighted text)

Line 262, as noted above, the current experiments do not exclude Barr recruitment. This needs to be measured.

Response authors: we agree with this reviewer, our experiments don't exclude barr recruitment. In fact, the opposite is the case. Our experiments were done based on the notion that **G proteins** engaged and activated by individual receptors are in the **OFF mode** (based on lack of signaling studied with canonical methods and FRET biosensors based on G protein activation), while **arrestins are still recruited**. Thus, the last figure addressed the search for a functional correlate of arrestin recruitment in the absence of active G proteins. We chose conditions for each receptor that show inactivity of cognate G proteins but recruitment of arrestin to examine whether receptors still internalize. Clearly, barr recruitment takes place for all receptors when G proteins are inactive (shown for DP2, GPR17, and FFA2 in Fig.1; for M3D in Fig. S8). So yes, **barr recruitment has been measured**, and the first statement of this paragraph has been rephrased to clarify that we wanted to monitor a functional correlate of barr recruitment when G proteins are OFF (see page 9, beginning of second paragraph).

Lines 267-268, as noted above, treatment with PTX does not "eliminate" all Gi/o, unless they actually do experiments to demonstrate this.

Response authors: we thank the reviewer for this comment and replaced "eliminated" by "inhibited and/or eliminated" because we used PTX+FR for DP2 and GPR17 but required Δ G12/13KO cells + PTX+FR for FFA2 (page 9, line 7 from the bottom of the page).

Line 273 – the reported data do not establish that “G protein-independent” receptor internalisation occurs – though it does speak to effects when Ga-dependent activation is inhibited.

Response authors: thank you for noting this inaccuracy. We now use the term “internalization when G protein activation was inhibited”.

Discussion – as noted above – the “G protein-independent” concept is not one that is universal and has been promoted by groups with a particular interest in arrestin-dependent signalling. Similarly, the current study is not “zero G” (unless this is specifically established). DMR is not “unbiased” (see note above). Elimination of G proteins and inactivation are not the same.

Response authors: “zero G” has been replaced by “zero functional G” throughout the manuscript; DMR is no longer referred to as unbiased; elimination and inactivation are used where appropriate (all replacements are boxed yellow).

Lines 312-313, is there any evidence that inactivation of Gq/11 by the inhibitors prevents recruitment of G protein. Also, when doing G protein recruitment experiments, this should be done as both GPCR to Galpha and GPCR to Gbg (as these are not equivalent when measured).

Response authors: we thank the reviewer for raising this issue. As we have not measured recruitment of PTX-treated Gi or FR-treated Gq to ligand-activated receptors, we followed the advice of this reviewer and replaced ‘zero G’ by ‘zero functional G’. We agree that it would be a very interesting follow-up project to further define ‘zero functional G’ and to perform studies such as those suggested above.

Line 320, it is not clear to this reviewer why they find it “puzzling” that SII is applied to and referred to as arrestin-biased. The literature is littered with investigators who do not properly understand concepts and quantification of biased agonism (I would include the current authors based on some of the statements in the current paper).

Response authors: we have replaced “puzzling” by “unfortunately” because we are surprised that SII is treated in the literature by some as **perfectly biased towards arrestin** although it is not. We have rephrased this section to clarify that we understand the conditional nature of the term bias (page 11, third paragraph, yellow text).

Line 324, correctly notes that all data are in HEK 293 cells. In this context, it would be appropriate to discuss some of the background-dependent variables that are at play, including levels of GRKs, lipid rafts, etc.

Response authors: we have dedicated a short section to background-dependent variables (please see page 11, last paragraph).

Lines 328-329, do the authors really mean “arisen” or “proposed”, the latter would be a more appropriate descriptor.

Response authors: “arisen” was replaced with “proposed” as suggested (page 11, last line).

Lines 333-337, there is speculation about G protein-independent binding of arrestin and implications

from structures in complex with receptor. In the limited studies with full receptor, the “pendant” binding that occurred was in presence of G protein. This section is overly speculative, in the view of this reviewer.

Response authors: we have deleted this overly speculative paragraph from the discussion.

Line 340, implies that “G protein-independent recruitment” of Barr has been established in the current work, whereas this is not true.

Response authors: we agree with the reviewer. The new wording ‘arrestin recruitment in the absence of active G proteins’ or ‘barr recruitment at zero functional G’ is **much better supported** by our experimental data. We used this expression as it is widely found in the scientific literature and adopted the term without questioning its accuracy.

Lines 345-346, while I agree that the gene deletion cell lines are an important tool, it needs to be noted that no fully G protein ablated cell line has been created (and indeed this may not be tolerated by cells). As such, there are still the limitations discussed above that need to be addressed.

Response authors: To address this concern, the term “collective absence of G proteins” has been replaced with “collective absence of **functional** G proteins”. Furthermore, we added to the Discussion that no fully G protein ablated cell line has been created (please see a new section on this topic on page 11, top paragraph).

In all pERK experiments, data is presented as fold-over-basal. In order for this to be meaningful, actual basal levels need to be reported.

Response authors: We agree with the reviewer. Presenting data as fold over basal may become problematic if basal levels show significant variations. This was not the case in our study. Basal ERK levels varied to some extent between cell lines and inhibitor treatments, yet were in a very similar range to calculate fold over basal with great confidence. Please see below the distribution of basal values for the DP2, the first receptor shown in Fig.3 of the main manuscript. We have not prepared Figures or Tables with basal values (the paper contains so much information already) but would be happy to do so upon specific request by this referee.

Under “Imaging” (p16), in these experiments PTX treatment is only for 1h. Under this condition, the effect of PTX is likely to be one reflected as activation of Gi. It is the O/N treatment that leads to depletion of the activateable pool. Why did the authors perform these experiments??

Response authors: thank you for making us aware of this typing error; PTX treatment was for 18 h. This is now corrected.

The reference list is not uniformly formatted.

Response authors: the reference list is now uniformly formatted.

In figures, DMR traces are “representative” but also apparently mean+SEM of 3 independent experiments (lines are all solid, and it is unclear what this means, or why, in this format only +SEM would be displayed).

Response authors: we have added a statement to the DMR method section to explain that each trace represents triplicate technical replicates over time presented as mean + SEM. Each experiment is repeated at least three times to obtain independent biological replicates.

Throughout the figure legends there is no detail on the specific measures that statistical tests are applied to.

Response authors: we are now more specific in all figure legends where statistics is applied (Fig. 1, Fig. S4, Fig. S6, Fig. S9).

For DMR experiments, quantitative data is presented for only a limited subset of data (even where concentration-response curves have been performed). Where is the quantitative mean data?

Response authors:

- quantitative mean data of Fig.1, panels A-F are in Fig.1, panels G-I;
- quantitative mean data of Fig.5, panels J-L are in Suppl. Fig. 10;
- quantitative mean data of Suppl. Fig.1, panels G,H are in Suppl. Fig.1, panel I;

Fig. S14 title, “Statistical analysis of.....” Where, exactly are these statistical analyses???

Response authors: we corrected the figure caption and replaced “statistical analysis” by concentration-effect curves...

Fig. S15, a band in the Barr2 blots is claimed to be non-specific. What is the evidence that supports this statement?

Response authors: As pointed out by this referee, the anti-beta-arrestin2 monoclonal antibody exhibited two immunoreactive bands in the parental HEK293 cells, and the higher MW bands remain in the beta-arrestin1/2-mutant clones. For the following reasons, we concluded that the lower MW (major) band corresponds to beta-arrestin2 and the higher MW (minor) band reflects non-specific reactivity of the antibody.

We used two distinct sets of sgRNAs to generate beta-arrestin1/2-mutant clones (ref. O’Hayre et al. Sci Signal 2017). CL1 and CL3 were generated by using one set of sgRNAs (5’-

TTCCCCGTGTCTTCGGGCC-3' for the ARRB1 gene and 5'-CCAAAAGCTGTACTACCATG-3' for the ARRB2 gene). CL2 was established by another set of sgRNAs (5'-CGCCTCCGCTATGGCCGGG-3' for the ARRB1 gene and 5'-TGACCGGTCCCTGCACCTCG-3' for the ARRB2 gene). All of the sgRNAs target deduced amino-acid region between the Finger loop and the Lariat loop. This region is encoded by all of the registered isoforms of beta-arrestin1 and 2 (<https://www.ncbi.nlm.nih.gov/gene/408>, <https://www.ncbi.nlm.nih.gov/gene/409>) and is critical for interaction with GPCRs. Genomic sequencing of the three clones (CL#1, CL#2 and CL#3) confirmed successful introduction of frameshift mutations in the targeted loci (ref. O'Hayre et al. Sci Signal 2017). Functional analyses clearly demonstrated that all three clones completely lacked ligand-induced GPCR internalization. Western blot for beta-arrestin1 showed no immunoreactive bands in the three clones (Fig. S11). Thus, we reasoned that the beta-arrestin2 functional protein is also depleted in the three clones and the remaining higher MW bands do not reflect residual beta-arrestin2 or cross-reactivity of beta-arrestin1, but rather are non-specific bands recognized by the antibody.

Reviewers' Comments:

Reviewer #1:

Remarks to the Author:

In the revised manuscript, authors have made a sincere attempt to address previously raised concerns. They have included additional data, expanded discussion and included additional references. One of the shortcomings that I still see is that they do not really provide compelling arguments and experimental data/analysis of the data to explain the substantial deviation from the reported literature. For example, Wisler et al., have shown that Carvedilol induced pERK is completely sensitive to barr1/2-KD (siRNA). However, it appears that authors here notice almost no difference in barr-KO cells (comparing panels 4a and b). As the authors point out, carvedilol experiment was never done in in "zero G" but why do the data on barr-KD differ so dramatically? SII also shows the same pattern i.e. no difference between WT and barr-KO (panels 4d and e)? Same is the case with V2R as well although there are three different types of reports with V2R in the literature (i.e. partial sensitivity to barr-KD, complete sensitivity to barr2-KD and enhancement upon barr2-KD). It is possible that more careful analysis of the existing data can help clarify this (e.g. if you compare either the maximal ERK response, or area under the curve, in panels 4a and b/4d and e/4g and h, do you see a statistically significant difference?). Overall, the notion that presence of functional G is essential for any component of ERK activation is now supported by the data in the manuscript. Unfortunately however, why ERK attenuation is observed upon barr-KD by several groups is not really tested experimentally or at least discussed and elaborated upon. The argument provided in the discussion about scaffolding ability of barrs appears to be weak and circular. I believe that extending the discussion further and more careful analysis of the data (statistical) should make the manuscript acceptable for publication. Almost every figure would benefit from a better statistical analysis of the data.

Reviewer #2:

Remarks to the Author:

The authors used HEK293 cells depleted of functional G proteins or both b-arrestins. The authors tested a wide range of class A GPCRs, including WT and mutant receptors, following stimulation by a variety of agonists that were reported to be unbiased or b-arrestin-biased. In addition to the levels of second messengers the authors tested the activation of ERK1/2, which was often presented as "G protein-independent b-arrestin-dependent" branch of signaling, and global mass redistribution (DMR), reflecting the changes in cell morphology regardless of the signaling pathway involved. In all cases the results were the same: in the absence of all functional G proteins there was no detectable signaling, whereas b-arrestins appeared to play a role in ERK activation downstream of G proteins, but not independently of them. Yet b-arrestin recruitment to active GPCRs, as well as agonist-induced internalization of these receptors appeared to occur in the absence of all functional G proteins.

Thus, this remarkably comprehensive study demonstrated that b-arrestin recruitment and receptor internalization does not require G protein activation, whereas every branch of signaling, including ERK1/2 activation, does. The novelty here lies in unambiguous approach: genetic or pharmacological ablation of G proteins and b-arrestins is complete, in contrast to partial siRNA knockdown employed in many previous studies. The conclusions, which are unexpected to many in the field, have wide implications in cell biology.

The manuscript was greatly improved in revision. The study became even more comprehensive and convincing with the inclusion of new data, clearer explanations, and improved referencing of earlier studies.

Minor editorial changes (that can be made in proofs) are needed: line 182, in view of the data presented here, it is better to use "were reported to also signal" instead of "thought to also signal"; line 313, the word "even" can be deleted (there is another "even" in the next sentence); line 319 "despite of the same" should read "despite the same" or "in spite of the same"; line 382,

"has" should be changed to "have" (the noun is plural, mechanisms); legend to Fig. S4, "Concentration-effect-curves" are usually called "dose-response curves" in pharmacology.

Reviewer #3:

Remarks to the Author:

The manuscript is significantly improved and has addressed most of major concerns.

Reviewers' comments:

Reviewer #1 (Remarks to the Author):

In the revised manuscript, authors have made a sincere attempt to address previously raised concerns. They have included additional data, expanded discussion and included additional references. One of the shortcomings that I still see is that they do not really provide compelling arguments and experimental data/analysis of the data to explain the substantial deviation from the reported literature. For example, Wisler et al., have shown that Carvedilol induced pERK is completely sensitive to barr1/2-KD (siRNA). However, it appears that authors here notice almost no difference in barr-KO cells (comparing panels 4a and b). As the authors point out, carvedilol experiment was never done in in “zero G” but why do the data on barr-KD differ so dramatically? SII also shows the same pattern i.e. no difference between WT and barr-KO (panel s 4d and e)? Same is the case with V2R as well although there are three different types of reports with V2R in the literature (i.e. partial sensitivity to barr-KD, complete sensitivity to barr2-KD and enhancement upon barr2-KD). It is possible that more careful analysis of the existing data can help clarify this (e.g. if you compare either the maximal ERK response, or area under the curve, in panels 4a and b/4d and e/4g and h, do you see a statistically significant difference?). Overall, the notion that presence of functional G is essential for any component of ERK activation is now supported by the data in the manuscript. Unfortunately however, why ERK attenuation is observed upon barr-KD by several groups is not really tested experimentally or at least discussed and elaborated upon. The argument provided in the discussion about scaffolding ability of barrs appears to be weak and circular. I believe that extending the discussion further and more careful analysis of the data (statistical) should make the manuscript acceptable for publication. Almost every figure would benefit from a better statistical analysis of the data.

Response authors: we thank this reviewer for the appreciation of the effort we made trying to further strengthen our manuscript with new data and expanded discussion. We are also grateful for accepting the notion that the presence of functional G protein is essential for any component of ERK activation. As **requested** we have now **further expanded the discussion** to elaborate on discrepancies for pERK data from different laboratories in barr-depleted cells. Discrepancies are indeed striking, not only for V2R as pointed out by this reviewer: for beta2AR **Wisler** found **pERK reduction** in partially barr-depleted cells by **71%** for **carvedilol** and by **42%** for **isoproterenol** (see Fig. 5A in Wisler et al., 2007; ref8 of our paper). In contrast, **O’Hayre** found **remarkable enhancement** by **181%** for **isoproterenol** (see Fig. 3F in O’Hayre et al., 2017; ref 29 of our paper). And finally **Lee** (ref 21 of our paper) found **no effect** for **Isoproterenol** in barr-depleted cells **exactly aligned with our findings** (Fig. 5a,b of our manuscript). **All three studies** used **partial depletion** of beta arrestins by siRNA or shRNA, yet **differ dramatically** concerning their qualitative conclusion.

Thus, **striking differences** are not only apparent **between our** and the **Wisler** study, **striking differences** are **equally apparent** for partial si/hRNA-mediated knockdown of barr1/2 for beta2AR, and for V2R exactly as noted by this reviewer. Regardless, we have expanded, as requested, the discussion around this issue to draw reader’s attention to the **big disparities in the field**, even when using the **very same experimental approach** such as partial reduction of barr expression by si/hRNAs. We have also offered two explanations potentially explaining such disparities in the discussion.

In addition, as **requested**, we provide what he/she refers to as more careful statistical analysis (comparison of results from the previous Figures 4a,b; d,e; g,h) in the Supplementary section as new Suppl. Figures 5 and 7. Yet, this careful statistical analysis does not change our very consistent overall picture: lack of arrestin does in no case lead to disappearance of ERK phosphorylation but may, as already observed for GPR17, be accompanied with reduced pERK1/2.

Statistical analysis of the remaining figures: We added statistical analysis back to the manuscript to selected panels that would benefit from such analysis: these are Figures 1,2,4,7,9 (please note that Figure numbering has been altered to adhere to the NCOMMS formal guidelines to fit display items + legend on one page). When absence or presence of responses is just too obvious (such as in Fig. 3g-i), and statistical claims are not made, statistics was not added back (advice of Dr. John Spouge, NIH statistician, spouge@ncbi.nlm.nih.gov; available for any request on revised statistics). Dr. Spouge recommends to **not use statistics** for datasets where the manuscript **text** does **not make claims on significance**, therefore we retained statistics mainly for those datasets where significance is either not obvious to the eye or is needed to support specific claims. We feel that this is also aligned with the recent comment in **Nature** from Nov28 “Five ways to fix statistics”.

Reviewer #2 (Remarks to the Author):

The authors used HEK293 cells depleted of functional G proteins or both b-arrestins. The authors tested a wide range of class A GPCRs, including WT and mutant receptors, following stimulation by a variety of agonists that were reported to be unbiased or b-arrestin-biased. In addition to the levels of second messengers the authors tested the activation of ERK1/2, which was often presented as “G protein-independent b-arrestin-dependent” branch of signaling, and global mass redistribution (DMR), reflecting the changes in cell morphology regardless of the signaling pathway involved. In all cases the results were the same: in the absence of all functional G proteins there was no detectable signaling, whereas b-arrestins appeared to play a role in ERK activation downstream of G proteins, but not independently of them. Yet b-arrestin recruitment to active GPCRs, as well as agonist-induced internalization of these receptors appeared to occur in the absence of all functional G proteins.

Thus, this remarkably comprehensive study demonstrated that b-arrestin recruitment and receptor internalization does not require G protein activation, whereas every branch of signaling, including ERK1/2 activation, does. The novelty here lies in unambiguous approach: genetic or pharmacological ablation of G proteins and b-arrestins is complete, in contrast to partial siRNA knockdown employed in many previous studies. The conclusions, which are unexpected to many in the field, have wide implications in cell biology.

The manuscript was greatly improved in revision. The study became even more comprehensive and convincing with the inclusion of new data, clearer explanations, and improved referencing of earlier studies.

Minor editorial changes (that can be made in proofs) are needed: line 182, in view of the data presented here, it is better to use “were reported to also signal” instead of “thought to also signal”; line 313, the word “even” can be deleted (there is another “even” in the next sentence); line 319 “despite of the same” should read “despite the same” or “in spite of the same”; line 382, “has”

should be changed to “have” (the noun is plural, mechanisms); legend to Fig. S4, “Concentration-effect-curves” are usually called “dose-response curves” in pharmacology.

Response authors: we thank this reviewer for his/her positive evaluation and the appreciation of our study. We have incorporated the suggested editorial changes and have highlighted these in yellow. Please note that I was taught by my pharmacology professor to use the term “dose-response” for a dose of a drug that is given to an animal whereas I should use concentration-effect curve if I am applying a certain concentration of ligand to cells. I have always been very careful to distinguish dose-response from concentration-effect, and would – for this particular distinction - recommend to keep the term ‘concentration-effect curve’ in the legend to Supplementary Figure 4.

Reviewer #3 (Remarks to the Author):

The manuscript is significantly improved and has addressed most of major concerns.

Response authors: we thank this reviewer for his/her positive evaluation.